# Coordinated control of senescence by lncRNA and a novel T-box3 co-repressor complex

**Pavan Kumar P[1,2], Uchenna Emechebe[3†], Richard Smith[4†], Sarah Franklin[5,6], Barry Moore[7], Mark Yandell[7], Stephen L Lessnick[2,4,8], Anne M Moon[1,2,7]\***

[1]Weis Center for Research, Geisinger Clinic, Danville, United States; [2]Department of Pediatrics, University of Utah, Salt Lake City, United States; [3]Department of Neurobiology and Anatomy, University of Utah, Salt Lake City, United States; [4]The Centre for Children's Cancer Research, Huntsman Cancer Institute, University of Utah, Salt Lake City, United States; [5]Nora Eccles Harrison Cardiovascular Research and Training Institute, University of Utah, Salt Lake City, United States; [6]Department of Internal Medicine, University of Utah, Salt Lake City, United States; [7]Department of Human Genetics, University of Utah, Salt Lake City, United States; [8]Department of Oncological Sciences, Huntsman Cancer Institute, University of Utah, Salt Lake City, United States

**\*For correspondence:**
ammoon@geisinger.edu

[†]These authors contributed equally to this work

**Competing interests:** The authors declare that no competing interests exist.

**Reviewing editor**: Michael R Green, Howard Hughes Medical Institute, University of Massachusetts Medical School, United States

**Abstract** Cellular senescence is a crucial tumor suppressor mechanism. We discovered a CAPERα/TBX3 repressor complex required to prevent senescence in primary cells and mouse embryos. Critical, previously unknown roles for CAPERα in controlling cell proliferation are manifest in an obligatory interaction with TBX3 to regulate chromatin structure and repress transcription of *CDKN2A-p16^INK* and the RB pathway. The lncRNA *UCA1* is a direct target of CAPERα/TBX3 repression whose overexpression is sufficient to induce senescence. In proliferating cells, we found that hnRNPA1 binds and destabilizes *CDKN2A-p16^INK* mRNA whereas during senescence, *UCA1* sequesters hnRNPA1 and thus stabilizes *CDKN2A-p16^INK*. Thus CAPERα/TBX3 and *UCA1* constitute a coordinated, reinforcing mechanism to regulate both *CDKN2A-p16^INK* transcription and mRNA stability. Dissociation of the CAPERα/TBX3 co-repressor during oncogenic stress activates *UCA1*, revealing a novel mechanism for oncogene-induced senescence. Our elucidation of CAPERα and *UCA1* functions in vivo provides new insights into senescence induction, and the oncogenic and developmental properties of TBX3.

## Introduction

Senescence is defined as irreversible arrest of cell growth and loss of replicative capacity (*Hayflick, 1965*). Senescent cells have a large, flattened morphology and a characteristic secretory phenotype. They may be multinucleate, exhibit nuclear distortion, and contain senescence-associated heterochromatin foci (SAHFs) (*Kosar et al., 2011*). Senescence can be induced by various stimuli such as DNA damage, metabolic or oxidative stress, or expression of oncoproteins (*Larsson, 2005*; *Kuilman et al., 2010*; *Coppé et al., 2011*).

The p16/retinoblastoma protein (RB) and p53 tumor suppressor pathways are key regulators of senescence induction and maintenance in many cell types (*Narita et al., 2003*). p14^ARF-p53 activates p21, whereas the p16^INK4a-RB pathway culminates in E2F transcriptional target repression and senescence (*DeGregori, 2004*). Expression of *CDKN2A-p14^ARF* and *CDKN1A-p21^CIP* is repressed by the related transcription factors TBX2 and TBX3; this is the postulated mechanism for senescence bypass

**eLife digest** Cell division and growth are essential for survival. But it is equally important that cells can stop dividing, because failing to do so can lead to the uncontrolled tumor growth seen in cancer. One such quality control mechanism is called senescence, which stops the growth and multiplication of cells that are old, damaged or behaving in ways that may harm the organism. All cells eventually stop dividing and undergo senescence, but a number of factors may trigger the process early, such as DNA damage, stress or the appearance of cancer-causing proteins.

Senescence can be harmful if it occurs too early in life and interferes with normal growth. Severe birth defects—including fatal heart problems and limb malformations—occur if senescence is inappropriately triggered early in development. Mutations in a gene encoding a protein called TBX3 have been linked to these severe birth defects.

Normally, TBX3 stops the production of other proteins that trigger senescence in early development, and helps to maintain stable conditions in adult cells. Understanding how it does so could help scientists understand normal cell function and aging, and also help to find ways to trigger senescence in cancerous cells.

Kumar et al. found that a protein called CAPERα—for short Coactivator of AP1 and Estrogen Receptor—forms a complex with TBX3 that stops cells dividing in living organisms in at least two different ways. One way is by altering how DNA is folded. The other way involves a non-coding strand of RNA from a gene called UCA1: this RNA prevents the degradation of proteins that stop cell division.

In normal proliferating cells, the CAPERα/TBX3 protein complex prevents the production of UCA1 RNA. In contrast, in cells that received a cancer causing stimulus, TBX3 and CAPERα physically separate: this activates production of UCA1 RNA and causes senescence. Further studies will be required to establish exactly how the CAPERα/TBX3 protein complex interacts with DNA and RNA to control senescence and prevent cancer.

of *Bmi1−/−* and SV40 transformed mouse embryonic fibroblasts by overexpressed TBX2 and TBX3, respectively (*Jacobs et al., 2000*; *Brummelkamp et al., 2002*; *Prince et al., 2004*).

Mutations in human *TBX3* cause a constellation of severe birth defects called ulnar-mammary syndrome (*Bamshad et al., 1997*). Efforts to understand the molecular biogenesis of this developmental disorder uncovered additional functions for TBX3 beyond transcriptional repression (*Fan et al., 2009*; *Frank et al., 2013*; *Kumar et al., 2014*) as well as critical roles in adult tissue homeostasis (*Frank et al., 2012*). The pleiotropic effects of TBX3 gain and loss of function suggest its molecular activities are context and cofactor dependent.

Despite the biologic importance of TBX3, few interacting proteins or target genes have been discovered, and the mechanisms underlying its regulation of cell fate, cell cycle, and carcinogenesis are obscure. We found that TBX3 associates with CAPERα (Coactivator of AP1 and Estrogen Receptor), a protein identified in a liver cirrhosis patient who developed hepatocellular carcinoma (*Imai et al., 1993*). CAPERα regulates hormone responsive expression and alternative splicing of minigene reporters in vitro (*Jung et al., 2002*; *Dowhan et al., 2005*) but its in vivo functions are unknown.

We show that a CAPERα/TBX3 repressor complex is required to prevent premature senescence of primary cells and regulates the activity of core senescence pathways in mouse embryos. We discovered co-regulated targets of this complex in vivo and during oncogene-induced senescence (OIS), including a novel tumor suppressor, the lncRNA *UCA1*. *UCA1* is sufficient to induce senescence and does so in part by sequestering hnRNP A1 to specifically stabilize *CDKN2A-p16^{INK}* mRNA. Our finding that CAPERα/TBX3 regulates p16 levels by dual, reinforcing mechanisms position CAPERα/TBX3 and *UCA1* upstream of multiple members of the p16/RB pathway in the regulatory hierarchy that controls cell proliferation, fate and senescence.

## Results

### CAPERα interacts with TBX3 in vivo

We recently discovered that TBX3 (human) and Tbx3 (mouse) interact with RNA-binding and splicing factors (*Kumar et al., 2014*). Among these, mass spectrometry of anti-TBX3 immunoprecipitated

(IP'd) proteins identified CAPERα (*Figure 1A*). Since TBX3 functions in mammary development and may contribute to the pathogenesis of breast and other hormone responsive cancers (*Douglas and Papaioannou, 2013*), its interaction with an ERα co-activator drove further investigation.

To determine if Tbx3 and Caperα interact in vivo, we IP'd endogenous Caperα from embryonic day (e)10.5 mouse embryo lysates (*Figure 1B*). Immunoblotting for Tbx3 confirmed its interaction with Caperα (*Figure 1C*, lane 5) and in vitro pull down assays revealed that their interaction is direct (*Figure 1D*, lane 6). *Caperα* is very broadly expressed during mouse embryonic development (Moon, unpublished), whereas *Tbx3* expression is very tissue specific and dynamic. We thus questioned whether the endogenous proteins interact in mouse tissues relevant to malformations seen in humans with UMS. Immunohistochemistry on sectioned e10.5 embryos showed that Tbx3 and Caperα proteins are co-expressed and have distinct localization patterns in different tissues: Caperα is detected in all dorsal root ganglia nuclei (*Figure 1E*), some of which contain co-localized Tbx3; in proximal limb mesenchyme, Tbx3 and Caperα co-localize in nuclei (*Figure 1F*) while in some distal cells and the ectoderm, Caperα is nuclear and Tbx3 is cytoplasmic (*Figure 1G*, white arrowheads). Such tissue specificity suggests that functions of the Caperα/Tbx3 complex are context dependent.

TBX3 DNA binding and repressor domains (DBD, RD) independently mediate interactions with partner proteins (*Carlson et al., 2001*; *Coll et al., 2002*; *Kumar et al., 2014*). To identify domains required for CAPERα interaction, we used a series of overexpression plasmids encoding mouse Tbx3 proteins with different mutations and functional domains (*Figure 1H*). The DBD, deleted repressor domain (ΔRD) and exon7 missense mutants are untagged proteins, whereas the C-terminal deletion mutants are Myc-tagged.

To assay the interactions of the untagged exogenous proteins with endogenous CAPERα in HEK293 cells, we needed to knockdown endogenous TBX3 with shRNA (*Figure 1I*). We previously demonstrated that mutant Tbx3 proteins produced from the overexpression plasmids are present in *TBX3* knockdown HEK293 cells (Figure 2 in *Kumar et al. 2014*). CAPERα is present and can be IP'd in the context of knockdown of endogenous *TBX3* and subsequent overexpression of mutant mouse Tbx3 proteins (*Figure 1J*). Immunoblot of anti-CAPERα IP'd samples shows that the endogenous CAPERα interacts with Tbx3 DBD mutant proteins (*Figure 1J'*, lanes 2 and 3 are L143P and N227D, respectively).

The Tbx3 deletion constructs encode Myc- tagged mutants that can be distinguished from endogenous TBX3, so interactions were assayed in wild-type HEK293 cells. Myc-tagged deletion mutants are IP'd by the anti-Myc antibody (*Figure 1K*), and probing anti-Myc IP'd material for CAPERα reveals that deletions more proximal than amino acid 655 disrupt the CAPERα/Tbx3 interaction (*Figure 1K'*).

The observation that deletions of the Tbx3 C-terminus disrupt the CAPERα/Tbx3 interaction led us to test whether the C-terminal repressor domain, which is crucial for the ability of Tbx3 to function as a transcriptional repressor and immortalize fibroblasts (*Carlson et al., 2001*), plays a role. Although the untagged ΔRD mutant is produced in TBX3 shRNA knockdown cells and IP'd by the anti-Tbx3 antibody (*Figure 1L* and *Kumar et al., 2014*) it does not interact with CAPERα (*Figure 1L'*). CAPERα also fails to interact with a C-terminal Tbx3 frameshift mutant similar to one identified in humans with UMS (*Bamshad et al., 1999*) (*Figure 1—figure supplement 1*).

## CAPERα and TBX3 are required to prevent premature senescence of primary human and mouse cells

Roles for TBX3 in cell cycle regulation and senescence of primary cells have not been reported. We employed loss-of-function to test whether TBX3 is required for sustained proliferation of primary cultured human foreskin fibroblasts (HFFs) and to determine if CAPERα functions in this process. We tested two different CAPERα and TBX3 shRNAs (please see 'Materials and methods' for sequences and location in target mRNAs). Both CAPERα and TBX3 shRNAs effectively decreased the amount of CAPERα mRNA (*Figure 2—figure supplements 1A and 2A,B*). Knockdown of either protein resulted in a dramatic increase in senescence associated β-galatosidase activity (SA-βgal, *Figure 2A–D*; *Figure 2—figure supplements 1 and 2C–H*). This effect is specific because it occurs with two different shRNAs and is rescued by overexpression of CAPERα (*Figure 2—figure supplement 1B,E,G,H*) and Tbx3 (*Figure2—figure supplement 2B,E,G,H*). For all subsequent experiments, CAPERα shRNA 'A' and TBX3 shRNA 'A' were used to perform knockdown (KD) in HFFs (protein knockdowns are shown in *Figure 2—figure supplements 1 and 2*, I panels).

The effects of CAPERα and TBX3 KD on HFF cell growth, and SA-βgal activity suggest induction of premature senescence. Consistent with this, both KDs dramatically influenced nuclear structure,

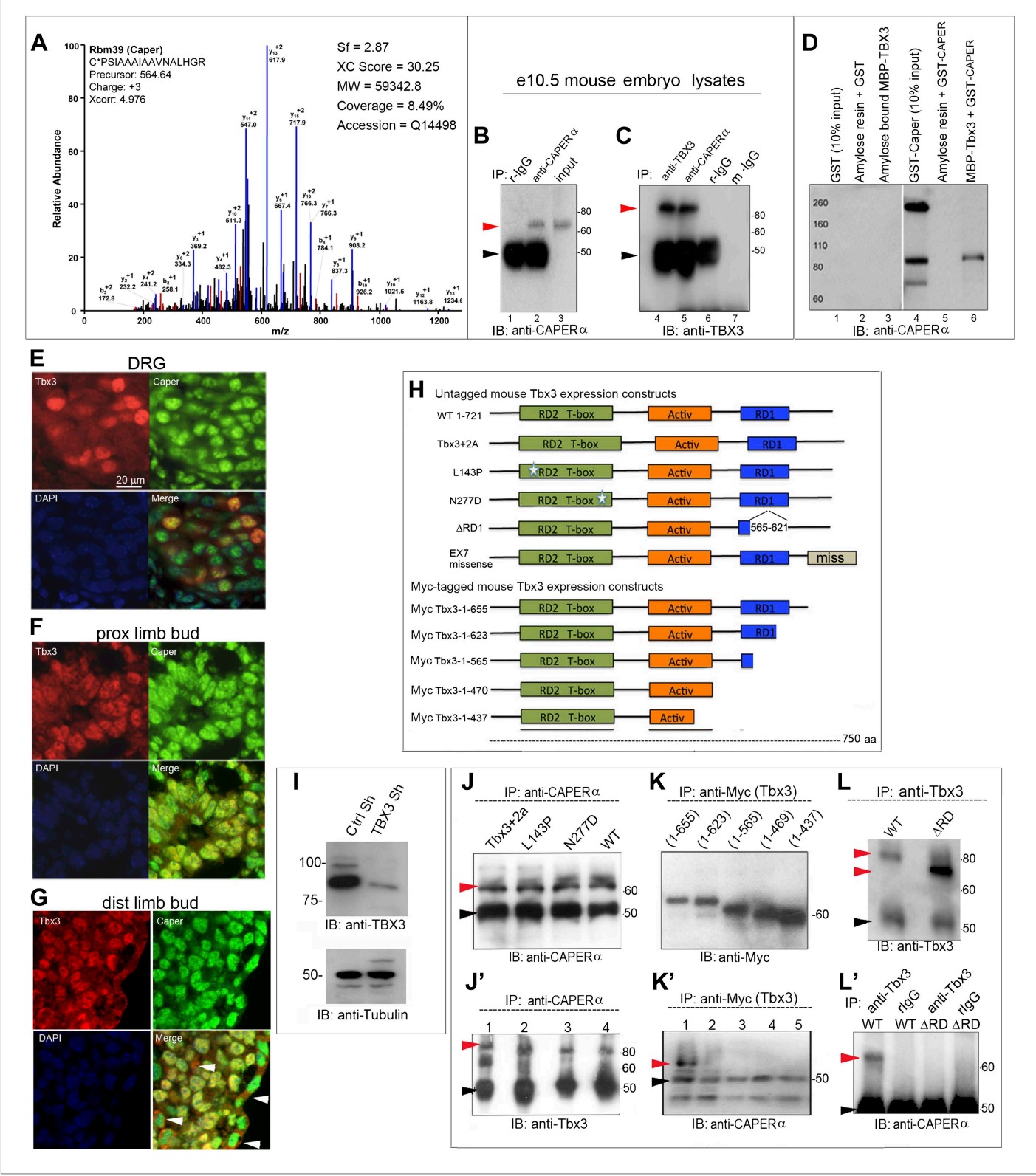

**Figure 1**. CAPERα and TBX3 directly interact via the TBX3 repressor domain. (**A**) Representative spectrum for CAPERα identified in anti-TBX3 co-IP of HEK293 cell lysates. Mass spec analysis identified six specific CAPERα peptides, providing 8.5% sequence coverage of the protein. This spectrum shows fragmentation of one of these peptides, C*PSIAAAIAAVNALHGR, with diagnostic b- and y-series ions shown in red and blue, respectively. * indicates

*Figure 1. Continued on next page*

*Figure 1. Continued*

carbamidomethylation. (**B**) Anti-CAPERα immunoblot (IB) analysis of anti-CAPERα immunoprecipitated (IP'd, lane 2) e10.5 mouse embryo lysates. Black arrowheads indicate IgG heavy chain and red indicate protein of interest (CAPERα or TBX3). (**C**) Anti-Tbx3 IB of anti-Tbx3 (lane 4) and anti-Caperα (lane 5) IP'd mouse embryo lysates. Rabbit (r)-IgG (lanes1, 6) and mouse (m)-IgG (lane 7) are negative controls. (**D**) In vitro MBP pull down assay: MBP and MBP-Tbx3 bound amylose affinity columns were incubated with GST or GST-CAPERα. Bound proteins were eluted, subjected to SDS-PAGE followed by IB with anti-CAPERα antibody. (**E–G**) Colocalization of Tbx3 and Caperα in vivo shown by immunohistochemical analysis of sectioned e10.5 mouse embryo: embryonic dorsal root ganglion (DRG, **E**), proximal (**F**), and distal (**G**) limb bud with anti-Tbx3 (red) and anti-Caperα (green) antibodies and DAPI (blue). White arrowheads in **G** label representative ectodermal and mesenchymal cells with cytoplasmic Tbx3 and nuclear Caperα. (**H**) Schematic representation of mouse Tbx3 overexpression constructs.Tbx3 DNA binding domain (DBD) point, ΔRD and exon7 missense proteins are untagged and the C-terminal deletion mutants are Myc-tagged. (**I**) Anti-TBX3 IB of HEK293 cell lysates transfected with control or anti-TBX3 shRNA. (**J**) Anti-CAPERα IB of anti-CAPERα IP'd samples from HEK293 cells transfected with anti-TBX3 shRNA and expressing mouse Tbx3 proteins listed at top. Production and IP of endogenous CAPERα is not affected by production of mutant Tbx3 proteins. (**J'**) Anti-Tbx3 IB of anti-CAPERα IP'd samples from HEK293 cells transfected with anti-TBX3 shRNA and expressing Tbx3 proteins as in **J**. The DBD point mutant proteins (lanes 2, 3) interact with CAPERα as efficiently as wild type Tbx3 (lanes 1, 4). (**K**) Anti-Myc IB of anti-Myc IP'd samples from HEK293 cell lysates expressing Myc-tagged mouse Tbx3 C-terminal deletion mutants. The mutant proteins are expressed and efficiently IP'd. These cells were not treated with anti-TBX3 shRNA because the expression constructs produce a Myc- tagged mutants that can be IP'd independently of endogenous TBX3. (**K'**) anti-CAPERα IB of anti-Myc IP'd samples from HEK293 cell lysates expressing Myc-tagged mouse Tbx3 C-terminal deletion mutants. These cells were not treated with anti-TBX3 shRNA because the expression constructs produce a Myc- tagged mutants that can be IP'd independently of endogenous TBX3. (**L**) Anti-Tbx3 IB of anti-Tbx3 IP'd samples from HEK293 cells transfected with anti-TBX3 shRNA and expressing wt or repressor domain deletion mutant (ΔRD) mouseTbx3. The shRNA does not prevent production of the overexpression proteins. (**L'**) Anti-CAPERα IB of HEK293 cells transfected with anti-TBX3 shRNA and expressing mouse wt or ΔRD Tbx3 proteins and IP'd with anti-Tbx3 or IgG. Loss of the repressor domain prevents interaction with CAPERα. Black arrowheads indicate IgG heavy chain and red indicate protein of interest (CAPERα or TBX3). TBX3, CAPERα = human; Tbx3, Caperα = mouse.

The following figure supplements are available for figure 1:

**Figure supplement 1**. Missense mutation of the C-terminus of Tbx3 disrupts interaction with CAPERα.

chromatin organization and formation of SAHFs (*Figure 2G–J*). Expression of senescence mediators was increased and conversely, expression of cell growth and cell cycle promoting genes was similarly decreased by CAPERα and TBX3 KD (*Figure 2K–M*). Increased expression of *CDKN2A-p16^INK* (henceforth referred to as *p16^INK*) and decreased *PCNA, E2F1* and *2, CDK2, CDK4, CDC2* transcripts indicate that CAPERα/TBX3 represses the p16/RB pathway in proliferating HFFs. *PMAIP1, CDKN1A-p21,* and other p53 pathway members were also increased. Collectively, these data indicate that CAPERα and TBX3 are required to prevent senescence of primary HFFs and act upstream of major cell cycle and senescence regulatory pathways.

### *Tbx3* null murine embryonic fibroblasts undergo p16/RB-mediated premature senescence, Caperα mislocalization and nuclear disruption

Tbx3 deficiency in mice causes lethal embryonic arrhythmias and limb defects however, these phenotypes are not due to increased apoptosis (*Frank et al., 2012* and Emechebe and Moon, unpublished). We hypothesized that Tbx3 may prevent senescence of embryonic cells, and so examined murine embryonic fibroblasts (MEFs) from e13.5 wild type (WT) and *Tbx3* null (−/−) embryos. WT MEFs undergo ~10 passages with regular, 20 hr doubling times. In contrast, *Tbx3*−/− MEFs had increased SA-βgal activity and ceased proliferating after only four passages (*Figure 2N–Q*). Most *Tbx3*−/− MEFs had distorted or ruptured nuclei (*Figure 2—figure supplement 3A–C*) and laminβ1 staining was already altered at passage 1 (*Figure 2—figure supplement 3B'*). *Caperα* null mutant embryos do not survive long enough to generate MEFs for complementary experiments (Emechebe and Moon, unpublished) however, Caperα localization is markedly abnormal in *Tbx3*−/− MEFS after only 1 passage (*Figure 2—figure supplement 3D–F'*). These data suggest that Tbx3 is required for preservation of nuclear architecture and to tether Caperα in its normal nuclear domains in proliferating cells.

Consistent with premature senescence seen in *Tbx3*−/− MEFs, key pro-senescence pathways are activated after loss of Tbx3 in vivo: in protein lysates from *Tbx3*−/− embryos, RB was hypophosphorylated on multiple serine residues, consistent with increased p16 and decreased Cdk2 and Cdk4 protein levels relative to control (*Figure 2R*). The levels of p21 and other senescence markers were increased, while numerous Cyclins and other Cdks were decreased (*Figure 2R*, *Figure 2—figure supplement 3G*). All of these findings are consistent with a requirement for Tbx3 to prevent senescence in embryonic mice and MEFs.

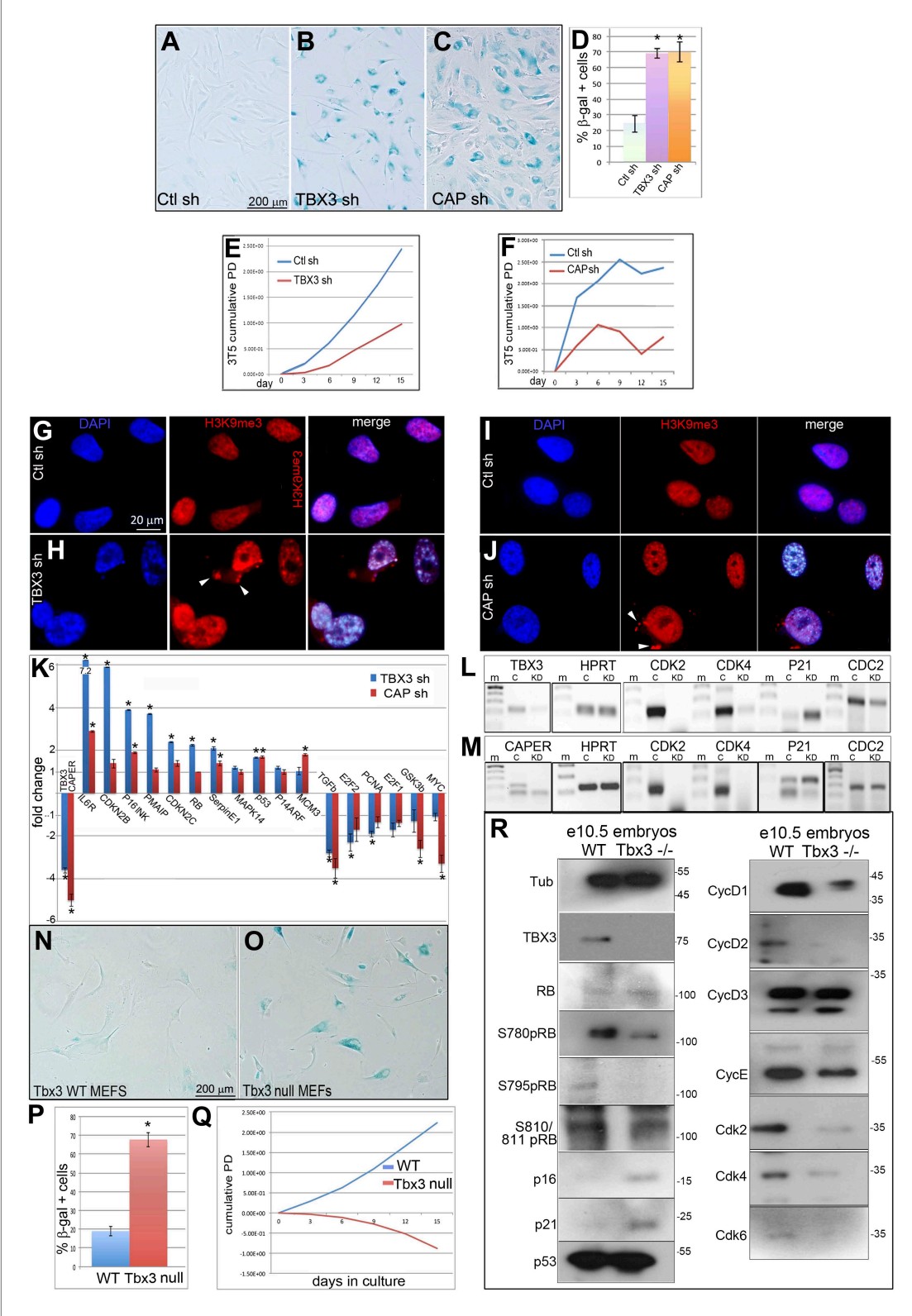

**Figure 2**. Knockdown of endogenous *CAPERα* and *TBX3* in primary human fibroblasts and mouse embryos induces premature senescence and disrupts expression of cell cycle and senescence regulators. (**A–C**) Representative bright field images of senescence associated β-galactosidase (SA-βG) assays of HFFs transduced with control, *TBX3* shRNA A or CAPERα shRNA A. Only occasional cells in the control transduction have detectable lacZ staining (blue)

*Figure 2. Continued on next page*

*Figure 2. Continued*

whereas knockdown of either TBX3 or CAPERα results in marked changes in cell morphology and increased lacZ staining. (**D**) Bar graph quantitating % beta-galactosidase positive cells from four replicate plates of SA-βgal assays. * indicates p<0.001 compared to control. (**E** and **F**) 3T5 cell proliferation assay (*Lessnick et al., 2002*) of cumulative population doublings in HFFs transduced at passage 30 with control, TBX3 or CAPERα shRNAs. These are representative curves of duplicate experiments; each point on the curve is a measurement of cell count from a single plating followed over the course of the experiment as described in methods. (**G–J**) Immunohistochemical analysis of H3K9me3 immunoreactivity (red) and DAPI (blue) in HFFs after knockdown with control (**G** and **I**), TBX3 (**H**), or *CAPERα* (**J**) shRNAs. Individual channels are shown and the merged image is on the right. Note increased nuclear punctate staining consistent with Senescence-associated heterochromatin foci (SAHFs) in both channels and evidence of nuclear disruption (white arrowheads in red channel) after loss of either TBX3 or *CAPERα*. (**K–M**) Analysis of cell cycle and senescence marker transcript levels in HFFs transduced with control, *TBX3,* or *CAPERα* shRNAs. (**K**) Relative transcript levels assessed by quantitative real time-PCR (qPCR) of cDNA. Values reflect fold change in knockdown HFFs relative to control after normalization to *HPRT* levels. Note general pattern of expression changes are similar in TBX3 (blue) and CAPERα (red) knockdowns. Data are plotted as fold change mean ± standard deviation. * indicates p<0.05 relative to control. (**L** and **M**) Agarose gel of PCR amplicons of cDNAs reverse transcribed from TBX3 (**L**) or CAPERα (**M**) shRNA knockdown HFF RNA reveals similar decreases in cell cycle promoting genes CDK2 and 4 in TBX3 and CAPERα knockdowns and increased p21 levels. (**N** and **O**) SA-βgal assay of wild type and *Tbx3* null MEFS reveals that Tbx3 is required to prevent premature senescence of primary murine embryonic fibroblasts (MEFs). (**P**) Quantitation of % beta-galactosidase positive cells from five replicate experiments exemplified in O, P. * indicates p<0.01. (**Q**) 3T5 cell proliferation assay of cumulative population doublings in wild-type and *Tbx3* null MEFs. These are representative curves from duplicate experiments; each point on the curve is a measurement of cell count from a single plating followed over the course of the experiment as described in 'Materials and methods'. (**R**) IBs to assay levels of cell cycle and senescence proteins in wild type and *Tbx3* null embryo lysates. Tubulin loading control is at top left (Tub). The changes at the protein level correlate with those observed at the RNA level (**K–M**) and RB is hypophosphorylated on multiple serine residues consistent with increased p16 and decreased CDK activity. TBX3, CAPERα = human; Tbx3, Caperα = mouse.

The following figure supplements are available for figure 2:

**Figure supplement 1**. Effective knockdown of endogenous CAPERα in primary human foreskin fibroblasts using viral shRNA transduction.

**Figure supplement 2**. Effective knockdown of endogenous TBX3 in primary human foreskin fibroblasts using viral shRNA transduction.

**Figure supplement 3**. *Tbx3* null murine embryonic fibroblasts (MEFS) have altered lamin β1 localization, nuclear disruption and mislocalized Caper*α*.

Previous studies have suggested that overexpression of TBX3 permits senescence bypass by directly repressing *CDKN2A-p14^ARF* (*p14^ARF*) to activate p53 (*Brummelkamp et al., 2002*), but a role for TBX3 in regulating *p16^INK* and the RB pathway has not been demonstrated. Thus, we expected that loss of p53 would rescue senescence resulting from TBX3 or CAPERα KD. To test this, we transduced TBX3 and CAPERα KD HFFs with shRNA to p53 (*Masutomi et al., 2003*) and assayed SA-βgal activity and growth. Surprisingly, although p53 shRNA effectively decreased p53 (*Figure 3—figure supplement 1A*), it did not rescue SA-βgal activity or growth arrest due to absence of TBX3 or CAPER*α* (*Figure 3B,E,G,H*). In contrast, shRNA-mediated KD of either RB (*Boehm et al., 2005*) or p16 (*Haga et al., 2007*) (*Figure 3—figure supplement 1B,C*) rescued these phenotypes in TBX3 and CAPERα KD cells (*Figure 3C,F–H,I–N*). These rescue experiments demonstrate that the p16/RB pathway mediates senescence downstream of CAPER*α* and TBX3 loss-of-function in primary cells.

## CAPERα/TBX3 regulates chromatin status of the *p16^INK* promoter

Increased p16 protein and RB hypophosphorylation in *Tbx3*−/− embryos and p16/RB-mediated senescence after CAPER*α* and TBX3 KD could result from loss of direct repression of *p16^INK* by CAPERα/TBX3 in proliferating cells. We screened 7 amplicons spanning ~6 kb upstream of *p16^INK* by ChIP-PCR of HFF chromatin (*Figure 3—figure supplement 2*); 3 amplicons were bound by CAPERα and TBX3 (*Figure 3O*, lanes 7, 10). Loss of either protein decreased the heterochromatic marks H3K9me3 (*Figure 3O*, lanes 14, 15) and H3K27me3 (*Figure 3—figure supplement 3*) and increased the euchromatic mark H3K4me3 (*Figure 3O*, lanes 17, 18). Notably, less CAPER*α* occupied *p16^INK* elements after TBX3 KD (*Figure 3O*, lanes 11) while the amount of TBX3 bound post-CAPER*α* KD was comparable to control (*Figure 3O*, lanes 9 vs 7). This is consistent with the abnormal localization of CAPER*α* seen in *Tbx3*−/− MEFS (*Figure 2—figure supplement 3D'–F'*) and indicates that CAPER*α* requires TBX3 to occupy *p16^INK* regulatory chromatin.

We examined whether CAPERα and/or TBX3 associate with promoters of other cell cycle genes that are transcriptionally dysregulated after CAPERα/TBX3 loss-of-function (*Figure 2K–M*). Antibodies against TBX3 and CAPER*α* ChIP'd the *p14^ARF* initiator (*Lingbeek et al., 2002*) (*Figure 3—figure supplement 4A*);

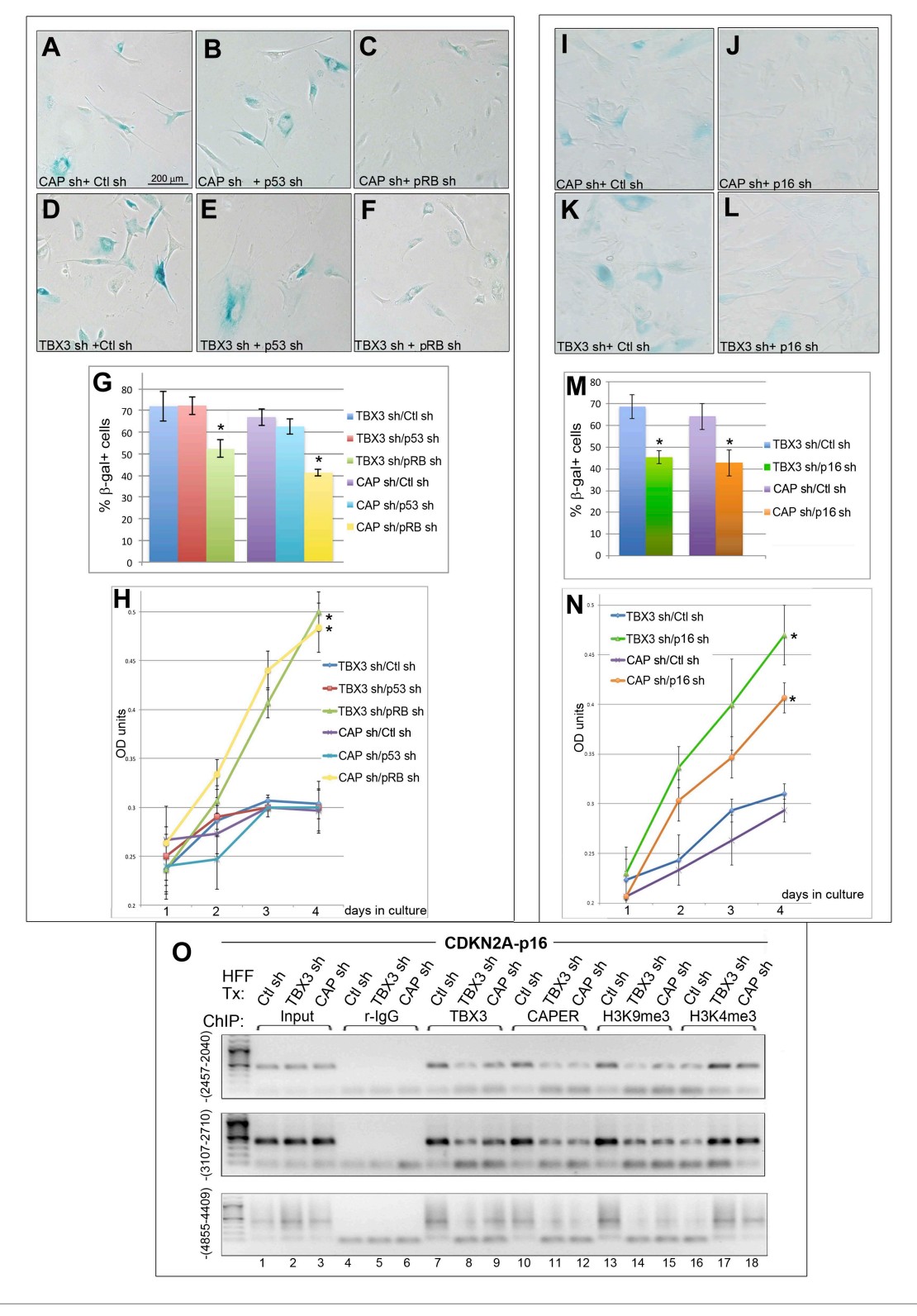

**Figure 3**. RB and p16 mediate senescence after CAPERα/TBX3 loss of function and CAPERα/TBX3 regulates chromatin structure of *CDKN2A-p16*. (A–F) SA-βgal assays of HFFs stably transduced with control (Ctl) or p53 (*Masutomi et al., 2003*) or RB (*Boehm et al., 2005*) shRNAs subsequently transduced with CAPERα or TBX3 shRNAs. (G) % Quantitation of **A–F** from three replicate experiments. * indicates p<0.05 relative to Control or p53 shRNAs. (H) Cell proliferation assayed by crystal violet incorporation (OD units) in HFFs treated as in **A–F**. * indicates p<0.001 relative to Ctl or p53 shRNAs. (I–L) SA-βgal

*Figure 3. Continued on next page*

*Figure 3. Continued*

assays of HFFs stably transduced with control or p16 (*Haga et al., 2007*) shRNAs subsequently transduced with CAPERα or TBX3 shRNAs. (**M**) % Quantitation of I-L from three replicate experiments. * indicates p<0.05 relative to Ctl shRNA. (**N**) Cell proliferation assayed by crystal violet incorporation (OD units) in HFFs treated as in I–L. * indicates p<0.01 relative to Ctl shRNA. (**O**) ChIP-PCR with antibodies listed at top on three regions upstream of the *CDKN2A-p16* transcriptional start site (TSS); position relative to (TSS) is indicated in parentheses at left of panels. PCR of input material used for the ChIP is shown under 'Input'. The shRNA transduced is listed above each lane (HFF Tx). TBX3 knockdown decreases binding of TBX3 (lanes 8) and CAPERα (lanes 11) to all three regions. CAPERα knockdown has minimal effect on TBX3 binding (lanes 9). Knockdown of either TBX3 or CAPERα decreases the repressive chromatin mark H3K9me3 (lanes14, 15) and increases the activating chromatin mark H3K4me3 (lanes 17, 18). TBX3, CAPERα = human; Tbx3, Caperα = mouse.

The following figure supplements are available for figure 3:

**Figure supplement 1**. Effective knockdown of p53, RB and p16 in HFFs.

**Figure supplement 2**. UCSC Genome Browser view of the *CDKN2A* locus and 5' regions screened for binding by CAPERα and TBX3.

**Figure supplement 3**. *CDKN2a-p16* H3K27 trimethylation markedly decreases in HFFS after knockdown of CAPERα or TBX3 consistent with activation of *CDKN2a-p16* expression.

**Figure supplement 4**. Testing CAPERα and TBX3 binding to *p14, p21, CDK2, CDK4,* and *CDKN1B* regulatory elements.

here too, TBX3 KD disrupted CAPERα binding (*Figure 3—figure supplement 4A*', red arrowhead). Neither CAPERα nor TBX3 associated with amplicons scanning 1.8 kb upstream of *CDKN1A-p21* or elements reportedly bound by TBX2 or TBX3 in other cell types (*Figure 3—figure supplement 4B*) (*Prince et al., 2004*; *Saramaki et al., 2006*; *Hoogaars et al., 2008*). Testing for association with known regulatory elements of *CDK2, CDK4, CDKN1B* was also negative (*Figure 3—figure supplement 4C–E*) (*Baksh et al., 2002*; *Wang et al., 2005*; *Louie et al., 2010*). These data indicate that in proliferating primary cells, CAPERα/TBX3 specifically and directly repress the *CDKN2A* locus by binding multiple regulatory sequence elements and regulating chromatin marks.

## Expression of the lncRNA *UCA1* is repressed by CAPERα/TBX3 and sufficient to drive senescence of primary cells

To identify novel genes repressed by CAPERα/TBX3, we employed differential display to detect transcripts that increased in response to KD of TBX3 and CAPERα in HEK293 cells (*Figure 4A–C*). Although most transcripts were unaffected by either KD, or changes were not shared (*Figure 4—figure supplement 1A*), *DUSP4* and *UCA1* were upregulated (*Figure 4D*, *Figure 4—figure supplement 1B*). DUSP4 is known to regulate cell survival and tumor progression, and overexpression induces senescence downstream of RB/E2F (*Torres et al., 2003*; *Wang et al., 2007*), thus placing CAPERα/TBX3 upstream of another p16/RB effector. Little is known about the function of the lncRNA *UCA1* (*Wang et al., 2006*, *2008*), so we investigated it further.

We found that shRNA KD of CAPERα or TBX3 in primary HFFS recapitulated the increase in *UCA1* transcripts seen in HEK293 cells (*Figure 4E–H*). We then tested whether CAPERα/TBX3 directly control transcription of *UCA1* by interacting with potential regulatory elements. Public ChIP data (http://genome.ucsc.edu/) indicate that the 2 kb upstream of *UCA1* may contain such elements. We assayed 3 amplicons in this region (*Figure 4I*: A1, A2, A3) by ChIP-PCR of TBX3 and CAPERα: only region A3 was bound (*Figure 4J,K*, lanes 18, red arrowheads).

We next determined whether increased *UCA1* expression in response to KD of CAPERα or TBX3 was associated with altered chromatin structure (as seen with *p16^INK^*, *Figure 3O*). *UCA1/A3* is normally in a heterochromatin configuration in HFFs, with repressive marks H3K9me3 and H3K27me3 (*Figure 4L*, lanes 12, 14) and little H3K4me3 (*Figure 4L*, lane 18). After TBX3 KD, activating chromatin marks replaced repressive ones (*Figure 4L*, lanes 13, 15 and 19) and markedly less CAPERα was bound (*Figure 4L*, lane 17, red arrowhead). CAPERα KD also led to loss of repressive marks on *UCA1/A3* (*Figure 4M* lanes 9, 16), although TBX3 remained bound (*Figure 4M*, lane 11, red arrowhead). Combined with previous findings, we conclude that: (1) TBX3 recruits CAPERα to *UCA1/A3* chromatin, (2) TBX3 alone is insufficient to repress *UCA1* and, (3) the default state of *UCA1* in proliferating HFFs is repression conferred by CAPERα/TBX3.

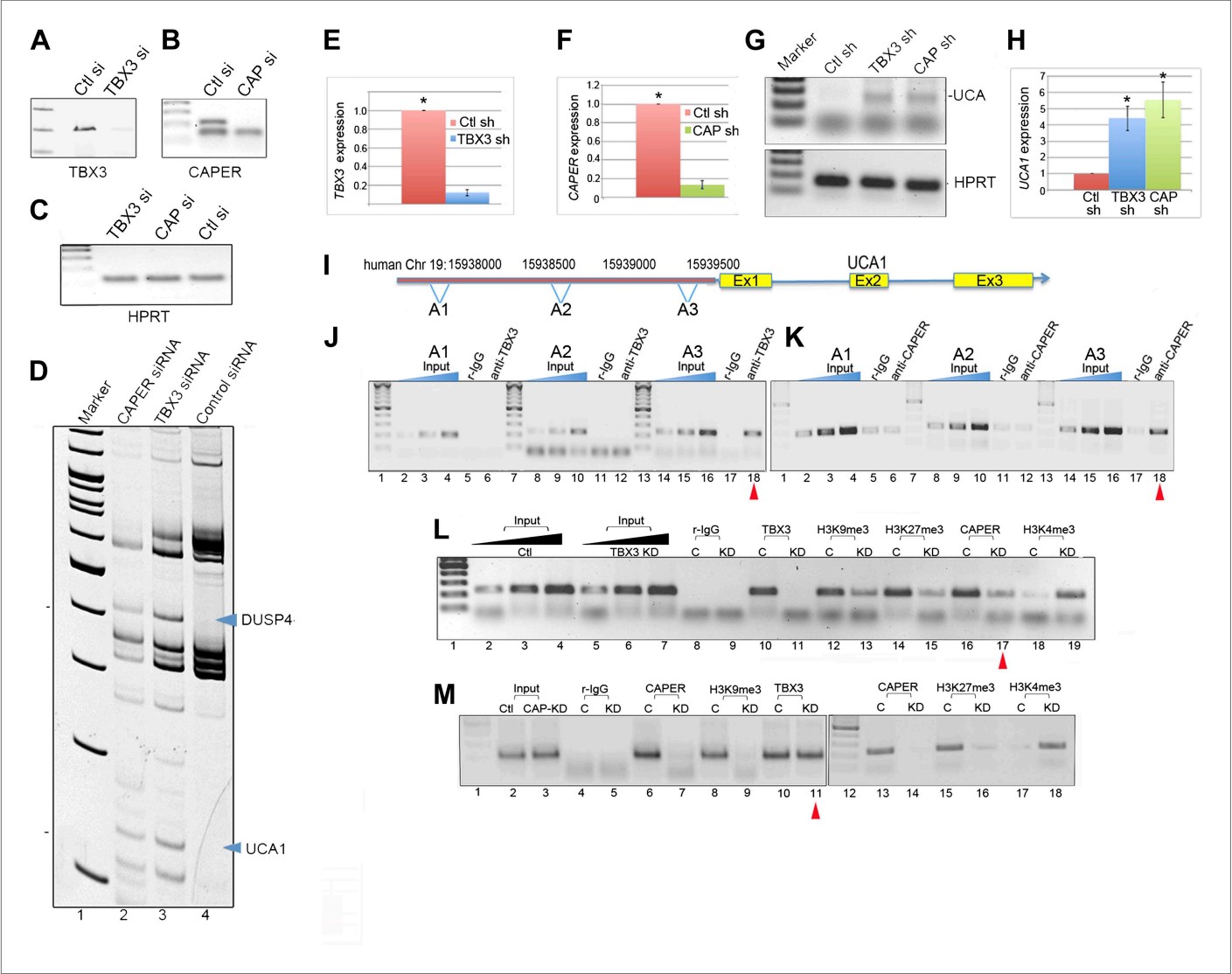

**Figure 4**. CAPERα/TBX3 directly represses expression of the long noncoding RNA *UCA1*. (**A–C**) Gel showing RT-PCR analysis of *TBX3, CAPERα,* and *HPRT* expression in control, TBX3 and CAPERα siRNA-transfected HEK293 cells. The siRNAs effectively decreased transcript levels of their targets. (**D**) Differential display: representative PAGE gel of cDNAs derived from random primed, RT-PCR'd mRNAs from CAPERα, TBX3 and control siRNA transfected HEK293 cells. Blue arrowheads denote upregulated transcripts subsequently identified by sequencing as *DUSP4* and *UCA1*. (**E** and **F**) qPCR analysis of *TBX3* and *CAPERα* transcript levels in control and *TBX3* or *CAPERα* shRNA transduced HFFs (repeat of experiment shown in ***Figure 2—figure supplements 1A and 2A***). (**G**) RT-PCR analysis of *UCA1* and *HPRT* gene expression in control, *TBX3* or *CAPERα* shRNA-transduced HFFs. (**H**) qPCR analysis of *UCA1* transcript levels in control, TBX3 or CAPERα shRNA transduced HFFs. Results confirm differential display result that KD of TBX3 or CAPERα results in increase in *UCA1* transcript levels. (**I**) Schematic representation of the *UCA1* locus with primer sets employed for ChIP-PCR amplification of denoted regions 5′ of gene (A1, A2, A3). (**J**) Anti-TBX3 ChIP-PCR of regions of the *UCA1* promoter in HFFs; only A3 is ChIP'd by TBX3 (lane 18, red arrowhead). (**K**) Anti-CAPERα ChIP-PCR of regions of the *UCA1* promoter in HFFs; only A3 chromatin is ChIP'd (lane 18, red arrowhead). (**L**) ChIP-PCR analysis of *UCA1*/A3 chromatin from in HFFs transduced with control (C) or TBX3 (KD) shRNA; ChIP antibodies are listed at top. Note decreased CAPERα binding after TBX3 KD (lane 17, red arrowhead), gain of activating mark H3K4me3 and loss of repressive marks H3K9me3 and H3K27me3. (**M**) ChIP-PCR analysis of *UCA1*/A3 with antibodies listed at top of panel in HFFs transduced with control (C) or *CAPERα* shRNAs. Note continued TBX3 binding despite *CAPERα* KD (lane 11, red arrowhead) and changes in chromatin marks parallel those seen in with TBX3 KD in panel **L**. TBX3, CAPERα = human; Tbx3, Caperα = mouse.

The following figure supplements are available for figure 4:

**Figure supplement 1**. Validation of differential display findings.

*UCA1* modulates behavior of bladder cancer cell lines (*Wang et al., 2008*), but there are no data on its function in primary cells; our results suggest that *UCA1* may be involved in premature senescence. *UCA1* transcripts are low in proliferating HFFs, but 4 days after overexpression of *UCA1* (*Figure 5A*), a robust SA-βgal response is evident (*Figure 5B–D*). Cells constitutively expressing *UCA1* ceased proliferating during selection and accumulated SAHFs (*Figure 5E,F*). Cell proliferation decreased in a *UCA1* dosage-sensitive manner (*Figure 5G–I*), consistent with reduced levels of cell cycle promoting transcripts and increased levels of pro-senescence ones (*Figure 5J*). These transcriptional changes were manifest at the protein level (*Figure 5—figure supplement 1*). Premature senescence resulting from overexpression of *UCA1* in HFFs reveals that this lncRNA is a novel regulator of cell proliferation and may function as a tumor suppressor in some contexts.

## Loss of *UCA1* delays the onset of oncogene-induced senescence

We tested the hypothesis that *UCA1* is required for induction of oncogene-induced senescence (OIS) in primary cells ('RAS': HFFs transduced with constitutively active $^{G12V}$RAS [*Serrano et al., 1997*]). There are markedly more *UCA1* transcripts in RAS compared to presenescent 'PS' HFFs (*Figure 5K*). Knockdown of *UCA1* in RAS HFFs reduced SA-βgal activity (*Figure 5L–Q*) and improved RAS cell growth: the number of Ki67 + RAS cells was increased at days 3 and 6 after *UCA1* KD (*Figure 5R*, P0 and P1). However, by passage 2, the number of Ki67 + cells was not statistically different in *UCA1* KD cells from control, despite persistently low levels of *UCA1* (*Figure 5S*) and decreased levels of pro-senescence transcripts (*Figure 5T*). Overall, this indicates that senescence can occur in the absence of high levels of UCA1 but that timely execution of the OIS program requires *UCA1*.

We next investigated whether increase in *UCA1* transcripts in OIS is a manifestation of loss of CAPERα/TBX3 occupancy/repression of *UCA1/A3*. Indeed, the repressor dissociates from *UCA1/A3* in RAS HFFs and *UCA1/A3* chromatin switches from heterochromatic to euchromatic marks (*Figure 5U*). This is consistent with the senescence-inducing effects of CAPERα/TBX3 loss-of-function (*Figure 2*) and resulting upregulation of *UCA1* (*Figure 4*), and establishes CAPERα/TBX3 regulation of *UCA1* in an independent model of senescence.

## *UCA1* promotes senescence by sequestering hnRNP A1 to stabilize *p16^{INK}* mRNA

Some lncRNAs influence transcription by recruiting chromatin modifiers to target genes (*Fatica and Bozzoni, 2014*). We tested whether the increased levels of prosenescence transcripts occurring in response to *UCA1* (*Figure 5J*) were the result activating chromatin changes however, ChIP-PCR assay for H3K9 acetylation of the *p16^{INK}*, *p14^{ARF}*, *CDKN1A-p21* (and other) promoters did not reveal changes in this activating mark in response to *UCA1* (*Figure 5—figure supplement 2*). We thus tested whether altered mRNA stability contributed to the observed changes. HFFs were transfected with *UCA1* expression or control plasmid and after 2 days, treated with Actinomycin D. Total RNA was collected at 0–4 hr post-treatment and mRNA levels assayed using RT-PCR. Remarkably, overexpression of *UCA1* resulted in the stabilization of mature *p16^{INK}*, *p14^{ARF}*, *E2F1*, and *TGFβ1* mRNAs: in the time frame examined, *p16^{INK}*, *p14^{ARF}*, and *E2F1* mRNAs do not decay and their $t_{1/2}$ values are therefore denoted as 'n' (no decay). The half-life estimates shown were calculated using linear regression; those best fit lines, their equations and R values are shown in *Figure 6—figure supplement 1*. $t_{1/2}$ of *p16^{INK}* mRNA in control cells was 3.9 hr vs n in *UCA1* overexpressing cells; *p14^{ARF}*, 2.4 vs n; *E2F1*, 7.2 vs n; *TGFβ1*, 1.9 vs 2.9. In marked contrast, *MYC*, *CDKN1A-p21*, *CDKN2D* and *RB* mRNAs decayed at rates indistinguishable from control (*Figure 6A*; *Figure 6—figure supplement 1*). The effects of *UCA1* overexpression on *p16^{INK}* mRNA stability were confirmed by Northern blot (*Figure 6—figure supplement 2*).

Regulation of *p16^{INK}* transcript stability is a critical mechanism for growth control (*Wang et al., 2005*; *Chang et al., 2010*; *Zhang et al., 2012*) and hnRNP A1 has been postulated to stabilize *p16^{INK}* mRNA (*Zhu et al., 2002*), but this has not been tested. To this end, we treated HFFs with siRNA to hnRNP A1 and used Actinomycin D to assess stability of *p16^{INK}* transcripts. Loss of hnRNP A1 (*Figure 6—figure supplement 3*) stabilized both *p16^{INK}* ($t_{1/2}$–2.1 in control vs 12.3 after HNRNP A1 knockdown) and *p14^{ARF}* mRNAs ($t_{1/2}$–1.5 in control vs 6.9 after hnRNP A1 knockdown) but not those of *E2F1* or *MYC* (*Figure 6B*). Half-life estimates were obtained as described for panel A and the best fit lines, their equations and R values are shown in *Figure 6—figure supplement 3B*. The differences in control half-lives between *Figure 6A,B* are likely attributable to the different treatments used: in A, control cells were transfected with pcDNA3.1 plasmid, while in B, control cells were transfected with

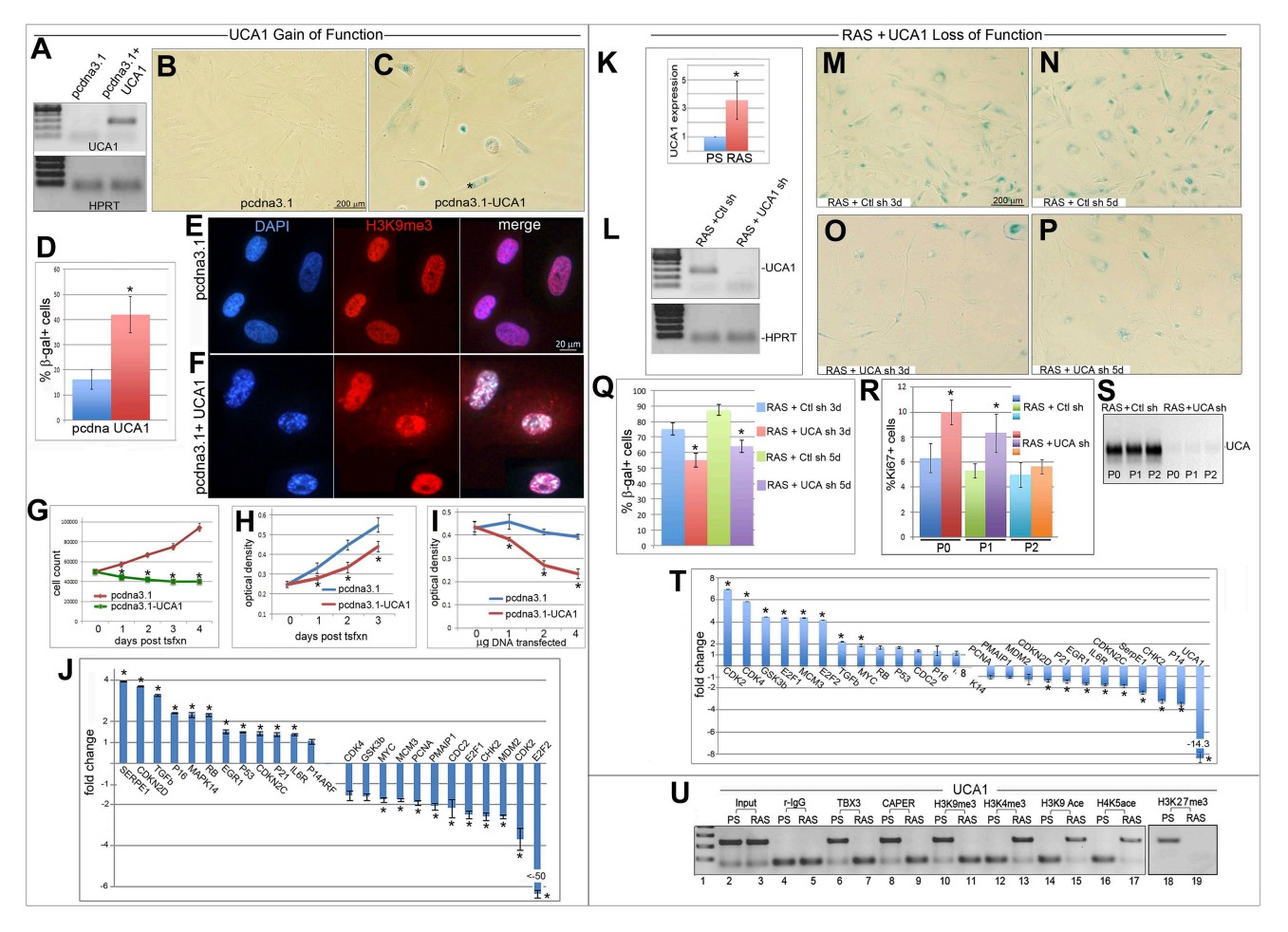

**Figure 5**. *UCA1* expression is sufficient to induce senescence and required for normal execution of oncogene-induced senescence. (**A**) *UCA1* and *HPRT* transcripts assessed by RT-PCR in control and *UCA1*-overexpressing HFFs. (**B** and **C**) Representative bright field images of SA-βgal assay of cultured HFFs transfected with control and *UCA1* overexpression plasmids. (**D**) % quantitation of SA-βgal cells from five replicates in control and *UCA1* overexpressing HFFs. * indicates p<0.05. (**E** and **F**) Immunohistochemical analysis reveals co-localization of H3K9me3 and DAPI in SAHFs in HFFs transfected with *UCA1* overexpression plasmid (**F**) but not control plasmid (**E**). (**G**) Cell count of control and *UCA1* overexpressing HFFs 3 days post transfection. Mean ± SD of 3 plates is shown at each time point. * indicates p<0.005 relative to control. (**H**) Crystal violet assay of cell growth in control and *UCA1* overexpressing HFFs transfected with 2 µg of expression or control vector and assayed daily for 3 days post- transfection. * indicates p<0.01 relative to control. (**I**) Crystal violet assay of HFFs cultured for 3 days after transfecting 0, 1, 2, or 4 µg of control or *UCA1* overexpression plasmid. * indicates p<0.01 relative to control. (**J**) Transcript levels assessed by qPCR; values reflect fold change in *UCA1*-overexpressing HFFs relative to control after normalization to *HPRT* levels. * indicates p<0.05 relative to control. (**K**) qPCR analysis of *UCA1* expression in untransduced, presenescent (PS) HFFs and HFFs transduced with constitutively active G12VRAS (RAS). * indicates p<0.05 relative to PS. (**L**) Efficient knockdown of *UCA1* transcripts in RAS HFFs with *UCA1* shRNA (quantitated in panel **T**). (**M–P**) SA-βgal assays of RAS HFFs transduced with either control or *UCA1* shRNA at 3 (**M** and **O**) and 5 (**N** and **P**) days post transduction. (**Q**) % quantitation of SA-βgal cells from six replicate experiments as represented in panels **M–P**. * indicates p<0.001 relative to control. (**R**) % quantitation of Ki67 + cells from three replicates in control vs *UCA1* shRNA transduced RAS HFFs. * indicates p<0.001 relative to control. (**S**) RT-PCR for *UCA1* transcripts shows persistent knockdown of *UCA1* in RAS shRNA cells with increasing passage (P0–P2). (**T**) qPCR analysis of fold changes in transcript levels of cell cycle and senescence genes after *UCA1* shRNA knockdown in RAS HFFs. * indicates p<0.05 relative to control. (**U**) ChIP-PCR analysis of *UCA1* region A3 with antibodies listed at top in PS and RAS HFFs. Note gain of activating (H3K4me3, H3K9ace, H4K5ace) and loss of repressive marks (H3K9me3, H3K27me3) at the *UCA1* locus after oncogene-induced senescence by RAS. TBX3, CAPERα = human; Tbx3, Caperα = mouse.

The following figure supplements are available for figure 5:

**Figure supplement 1**. Western blots showing changes in protein levels in response to *UCA1* overexpression in HFFs.

**Figure supplement 2**. ChIP-PCR assay for H3K9 acetylation of known regulatory elements of prosenescence and cell cycle genes whose expression is dysregulated after *UCA1* overexpression.

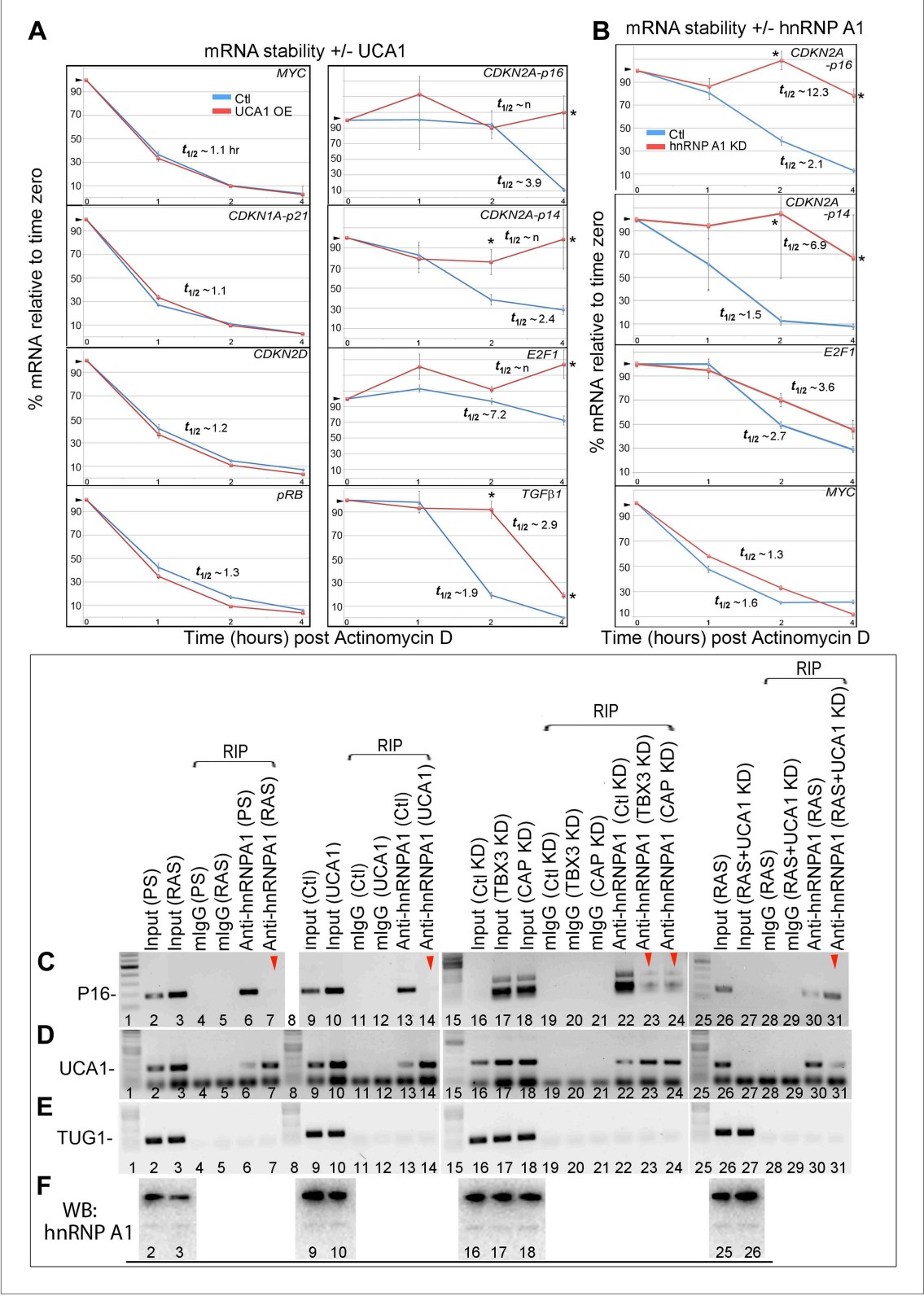

**Figure 6**. UCA1 stabilizes CDKN2A-p16 mRNA levels during senescence by sequestering hnRNP A1. (**A**) Graphs of transcript levels assayed by RT-qPCR in HFFs transfected with control (blue) or *UCA1* (red) expression plasmids and treated with Actinomycin (**D**). Y axis shows % mRNA level relative to time zero and X axis shows time in hours assayed post treatment. The estimated half-lives ($t_{1/2}$) were obtained using linear regression; the best fit lines, their

*Figure 6. Continued on next page*

*Figure 6. Continued*

equations and R values are shown in *Figure 6—figure supplement 1*. * indicates p<0.04 for *p16^INK* and p<0.01 for all others. (**B**) Assay of mRNA levels in HFFs transfected with control or hnRNPA1 siRNA and treated with Actinomycin D. Axes and $t_{1/2}$ calculations are as in panel **A**. * indicates p<0.05. (**C–E**) Agarose gels of RT-PCR products assessing levels of *CDKN2A-p16* (p16, panel **C**), *UCA1* (panel **D**), and negative control lncRNA *TUG1* (panel **E**) transcripts in PS and RAS HFFs treated as labeled at top and subjected to RIP with anti-hnRNPA1 antibody. mIgG lanes are negative controls for RIP assays. Gels from left to right show: PS vs RAS; control vs *UCA1* overexpression; control vs TBX3 or CAPERα knockdown; RAS vs RAS/*UCA1* knockdown. (**C**) Lane 7 (red arrowhead) shows loss of *p16^INK* /hnRNP A1 interaction in RAS. Lane 14 (red arrowhead) shows loss of *p16^INK* /hnRNP A1 interaction with *UCA1* overexpression. Lanes 23 and 24 show loss of *p16^INK* /hnRNP A1 interaction after TBX3 or CAPERα knockdown. Lane 27 shows that *UCA1* knockdown decreases the total amount of *p16^INK* mRNA in RAS cells. Lane 31 shows that *UCA1* knockdown increases *p16^INK* mRNA/hnRNP A1 binding (red arrowhead) in RAS cells, even though there is less total *p16^INK* (lane 27). (**F**) Panels show immunoblots to detect hnRNP A1 protein in input samples assayed in panels **C–E**. Lanes are numbered to correspond with panels above. TBX3, CAPERα = human; Tbx3, Caperα = mouse.

The following figure supplements are available for figure 6:

**Figure supplement 1**. Graphs showing best fit lines, their equations, and R values used to calculate estimated mRNA half-life values shown in *Figure 6A*.

**Figure supplement 2**. Northern blot assay of *p16^INK* mRNA levels in the absence and presence of *UCA1*.

**Figure supplement 3**. Graphs showing best fit lines, their equations and R values used to calculate estimated half-life values after hnRNP A1 siRNA knockdown shown in *Figure 6B*.

**Figure supplement 4**. RNA Immunoprecipitation analysis of hnRNP A1 interactions with *Myc* and *p14ARF* mRNAs.

**Figure supplement 5**. RIP-PCR of HFF lysates using antibodies listed at top.

**Figure supplement 6**. RIP-PCR indicates that *RB, p21,* and *CDK6* mRNAs do not interact with hnRNP A1 in PS or RAS HFFs.

control siRNA. The half-life of an mRNA is cell/context specific (as evident in the differences in control half-lives in 6A vs 6B) and in general, cell cycle regulatory genes have short half-lives (*Sharova et al., 2009*). The $t_{1/2}$ of *p16^INK* mRNA we observed in HFFs transfected with either control plasmid ($t_{1/2}$–3.9) or control siRNA ($t_{1/2}$–2.1) is similar to that reported in HeLa cells ($t_{1/2}$–2.9) (*Chang et al., 2010*). The results we obtained were also similar to those reported for *MYC* mRNA (*Herrick and Ross, 1994*; *Sharova et al., 2009*), *CDKN1A* mRNA in HT29-tsp53 cells (*Melanson et al., 2011*) and ES cells (*Sharova et al., 2009*), and *E2F1* mRNA in ES cells (*Sharova et al., 2009*). The half- lives of *Rb* and *TGFβ1* are mRNAs extremely variable and those we obtained in HFFs were shorter than reported in ES cells (*Sharova et al., 2009*).

We next used RNA-IP (RIP) to determine if hnRNP A1 binds *p16^INK* and *p14^ARF* mRNAs in proliferating cells and found that this was indeed the case (*Figure 6C*, lane 6 and *Figure 6—figure supplement 4*). Remarkably, hnRNP A1/*p16^INK* binding was lost in RAS HFFs (*Figure 6C*, lane 7), despite an overall increase in the number of *p16^INK* transcripts (*Figure 6C*, lane 3). As shown previously, *UCA1* RNA levels also increase with RAS (*Figure 6D*, lane 3). *UCA1* is bound by hnRNP A1 in PS cells (*Figure 6D*, lanes 6, 7; *Figure 6—figure supplement 5*), but unlike *p16^INK*, the hnRNP A1/*UCA1* interaction increases in RAS cells (*Figure 6D*, lane 7). *TUG1* lncRNA serves as a negative control (*Figure 6E*). Protein levels for hnRNP A1 are shown in *Figure 6F*. The interaction between *UCA1* and hnRNP A1 is specific, as *UCA1* does not bind hnRNP K, C1/C2, H, U, or D (*Figure 6—figure supplement 5*). Although hnRNP A1 binds *MYC* and *p14ARF* mRNAs (*Figure 6—figure supplement 4*), it does not bind *RB*, *p21* or *CDK6* mRNAs under the numerous conditions tested (*Figure 6—figure supplement 6*).

The opposite binding properties of *UCA1* and *p16^INK* mRNA with hnRNP A1 in PS vs RAS HFFs led us to postulate that *UCA1* stabilizes *p16^INK* mRNA during OIS by disrupting the interaction between hnRNP A1 and *p16^INK* mRNA. In control transfected proliferating cells, there is robust binding of *p16^INK* to hnRNP A1 (*Figure 6C*, lane13), but direct overexpression of *UCA1* (*Figure 6D*, lane 10) or that resulting from TBX3 or CAPERα KD (*Figure 6D*, lanes 17, 18) disrupts the hnRNP A1/*p16^INK* mRNA interaction (*Figure 6C*, lanes14, 23, 24, red arrowheads). These findings support the hypothesis that loss of hnRNP A1/*p16^INK* mRNA interaction in OIS (*Figure 6C*, lane 7) is the result of increased *UCA1* expression and its binding and sequestration of hnRNP A1 (*Figure 6D*, lane 7). To further test this, we used shRNA to KD *UCA1* in RAS HFFs (*Figure 6D*, lane 27). *UCA1* KD restored the interaction between

hnRNP A1 and *p16^INK* mRNA (*Figure 6C*, lane 31) and led to lower levels of total *p16^INK* mRNA (*Figure 6C*, lane 27), a finding consistent with the negative effects of hnRNP A1/ *p16^INK* interaction on stability of *p16^INK* transcripts. The effects of *UCA1* on *p16^INK* mRNA stability are specific, because hnRNP A1 inter-actions with *MYC* or *p14^ARF* mRNAs are unaffected by *UCA1* (*Figure 6—figure supplement 1*).

In total, these findings indicate that in proliferating cells, the very low quantity of *UCA1* transcripts is insufficient to disrupt hnRNP A1/*p16^INK* binding, and levels of *p16^INK* mRNA are low due to: (1) direct repression by CAPERα/TBX3 and, (2) *p16^INK* mRNA instability conferred by hnRNP A1. When *UCA1* levels increase during OIS, by *UCA1* overexpression, or via KD of CAPERα/TBX3, *UCA1* binds and sequesters hnRNP A1, preventing it from destabilizing *p16^INK* mRNA.

## The CAPERα/TBX3 co-repressor dissociates during oncogene-induced senescence leading to activation of *UCA1* and pro-senescence pathways

Increased p16 protein is required for RAS-induced senescence in MEFS and some human cell types (*Serrano et al., 1997*), leading us to determine whether OIS affects CAPERα/TBX3 occupancy of *p16^INK* chromatin. *CDKN2A-p16^INK* genomic regulatory elements bound in PS HFFs (*Figure 4I*) were not occu-pied by either TBX3 or CAPERα in RAS HFFs (*Figure 7A*). Chromatin marks on these regions switched from heterochromatic to euchromatic (*Figure 7B*, *Figure 7—figure supplement 1A*). This was also observed with *UCA1/A3* (*Figure 5U*) and *DUSP4* chromatin (*Figure 7—figure supplement 1B*).

We investigated the possibility that altered quantity of either CAPERα or TBX3 could disrupt the stoichiometry of their interaction and cause dissociation from *p16^INK* and *UCA1* regulatory elements in OIS. Surprisingly, both TBX3 and CAPERα protein levels were increased in RAS HFFs (*Figure 7C*), but they no longer co-IP'd (*Figure 7D*, red box). Immunocytochemistry of endogenous TBX3 and CAPERα in PS and RAS HFFs confirmed increased protein levels in OIS (*Figure 7F–M*), and revealed dramatic changes in CAPER*α* localization: CAPER*α* immunoreactivity became concentrated in large intranuclear foci (*Figure 7L,M*), as we previously observed in early passage *Tbx3−/−* MEFS (*Figure 2—figure sup-plement 2D–F*). These foci are distinct from SAHFs and PML bodies (*Figure 7M* and *Figure 7—figure supplement 2*).

To further investigate the molecular basis of senescence initiation after loss of CAPERα/TBX3, we performed genome-wide transcriptional profiling 2 days post CAPER*α*, TBX3 and control KD in HFFs. More than half of the transcripts with expression altered 1.5-fold or more by CAPER*α* KD were similarly affected by loss of TBX3 (N = 2375 CAPER*α* KD, 2188 TBX3 KD; 1157 co-regulated, p<<<<0.0001, *Figure 7—source data 1–3*, *Figure 7N,O*). Gene ontology-biologic process (GO-BP) analysis with DAVID (*Huang da et al., 2009a, 2009b*) showed highly significant co-regulation of 'transcription reg-ulation' (increased expression) and 'cell-cycle' (decreased expression) transcripts (*Figure 7N,O*). We tested a subset of these with known roles in senescence by qPCR: 100% validated and were similarly altered by RAS (*Figure 7—figure supplement 3*). Further interrogation of this group revealed that *IL6* and *HDAC9* are CAPERα/TBX3 direct targets and their upregulation in RAS is associated with loss of CAPERα/TBX3 binding (*Figure 7—figure supplement 4*).

We compared CAPERα/TBX3 co-regulated transcripts to a published data set comparing PS and ^G12V^RAS fibroblasts (*Loayza-Puch et al., 2013*). This revealed that 11% of CAPERα/TBX3 up-regulated transcripts were also increased by RAS (*Figure 7N'*); among these, GO-BP 'programmed cell death' (31%) and 'transcription regulation' (34%) were highly overrepresented. 30% of CAPERα/TBX3 down-regulated transcripts were also in the RAS data set; >1/3 of these were cell cycle genes (*Figure 7O'*). In all comparisons, the number of transcripts common to both groups was greater than predicted by chance and highly statistically significant (*Figure 7—source data 3*). KEGG pathway analyses revealed overrepresented pathways that were common to both CAPERα/TBX3 and RAS data sets (*Figure 7N–O'*, pie charts), but notably fewer pathways were shared in the upregulated group: JAK/STAT, TLR and TGFβ signaling pathways were only significantly overrepresented in the CAPERα/TBX3 data set.

## Discussion

Our knowledge of the regulatory mechanisms that govern the onset and maintenance of senescence in different contexts must be considered fragmentary (*Wang and Chang, 2011*; *Fatica and Bozzoni, 2014*). In this study, we provide compelling evidence for critical and novel functions of CAPERα, the lncRNA *UCA1* and TBX3 in the regulation of cell proliferation and senescence. We have discovered a CAPERα/TBX3 complex that is required to prevent senescence of primary human and mouse cells in

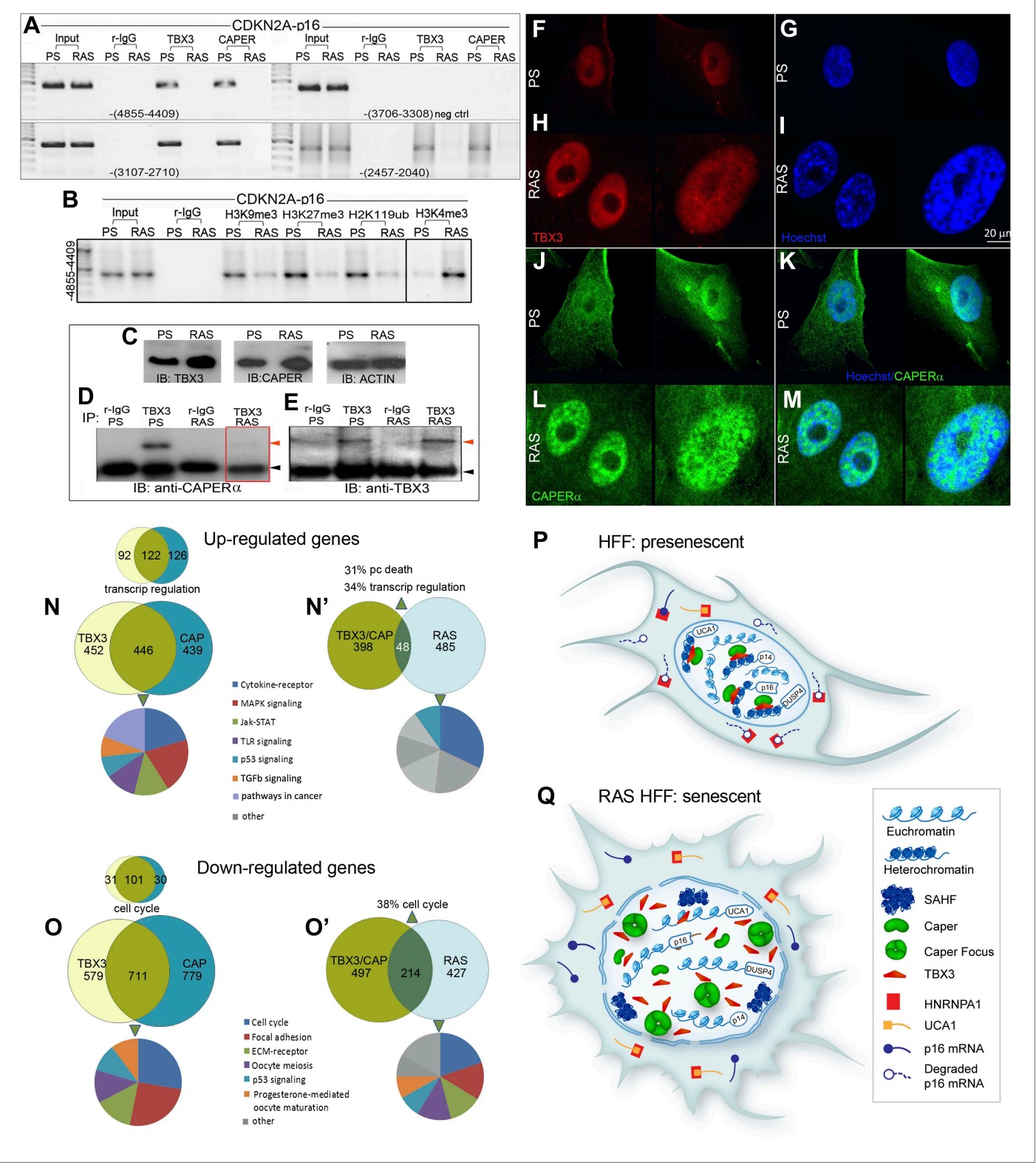

**Figure 7**. Disruption of the CAPERα/TBX3 repressor by OIS activates *CDKN2A-p16* and *UCA1* to trigger a senescence transcriptional response. (**A**) ChIP-PCR of regions upstream of the *CDKN2A-p16* transcriptional start site (position relative to TSS in parentheses) in PS and RAS HFFs; the −-3706–3308 amplicon is a negative control. OIS disrupts binding of *p16* regulatory elements (initially identified in *Figure 3O*) by TBX3 and CAPERα. (**B**) ChIP-PCR of *p16* -4855 element shown in **A**. Decreased TBX3 and CAPERα binding in RAS correlates with loss of repressive chromatin marks and gain of activating

*Figure 7. Continued*

marks. Evaluation of chromatin marks on the other *CDKN2A-p16* CAPERα/TBX3- responsive regulatory elements is shown in *Figure 7—figure supplement 1A*. (**C**) IBs for TBX3, CAPERα, and actin loading control show increased amount of both proteins in RAS compared to PS HFFs. (**D**) Anti-TBX3 and anti-CAPERα IBs of IP'd proteins from PS and RAS HFFs. (**F–M**) Immunocytochemical staining of PS (**F, G, J, K**) and RAS (**H, I, L, M**) HFFS for TBX3 (**F** and **H**), Hoechst (DNA; **G** and **I**), CAPERα (**J** and **L**). Panels **K** and **M** are merged Hoechst/CAPERα. Scale bar for all panels is sown at lower right of panel **I**. (**N–O′**) Functional analyses of genome wide transcriptional profiles of TBX3 KD, CAPERα KD, and control HFFs. All comparisons were statistically significant with p values <<<<0.0001; see *Figure 7—source data 3* for hypergeometric test, as implemented in the R statistical language, used to test significance of the number of genes found to be co-regulated between samples. (**N**) Venn diagrams show highly significant number of CAPERα/TBX3 co-upregulated transcripts (446 total), especially in the GO biologic process (BP) category of transcriptional regulation (122 transcripts) as assayed with DAVID. Pie chart shows KEGG pathway analysis of co-regulated genes. (**N′**) Venn diagram showing 48 CAPERα/TBX3 co-upregulated transcripts also upregulated by RAS/OIS (*Loayza-Puch et al., 2013*), especially in BP categories of transcriptional regulation and programmed cell (pc) death. qPCR validation of coregulated genes is in **S**. *Figure 6A*. Pie chart shows KEGG pathway analysis of OIS dataset. (**O** and **O′**) As in **N** and **N′** but for downregulated genes. Pie chart in **O′** shows KEGG pathway analysis of OIS data set; note most pathways are the same as in TBX3/CAPERα. (**P** and **Q**) Models of CAPERα/TBX3 repressor and *UCA1* function in proliferating (PS) HFFs vs RAS HFFs. In PS cell nuclei, CAPERα/TBX3 represses *UCA1, p16, p14,* and *DUSP4* promoters in heterochromatin which permits ongoing cell proliferation. RAS disrupts the CAPERα/TBX3 complex and CAPERα relocates to dense intranuclear foci. Pro-senescence genes including *UCA1* and *p16* are converted to euchromatin and their expression/products induce senescence. In the cytoplasm of PS cells, hnRNP A1 binds and destabilizes *p16* mRNA, but activation of *UCA1* expression in OIS allows *UCA1* to sequester hnRNP A1 and stabilize *p16* mRNA. TBX3, CAPERα = human; Tbx3, Caperα = mouse.

The following source data and figure supplements are available for figure 7:

**Source data 1**. Differentially expressed genes after knockdown of CAPERα in HFFs detected by RNA-Seq.

**Source data 2**. Differentially expressed genes after knockdown of TBX3 in HFFs detected by RNA-Seq.

**Source data 3**. Determining the statistical significance of shared differentially expressed genes using the hypergeometric test, as implemented in the R statistical language (phyper).

**Figure supplement 1**. Repression of *CDKN2A-p16* and *DUSP4* by CAPERα /TBX3 correlates with chromatin architecture and is relieved during onco-gene induced senescence.

**Figure supplement 2**. CAPERα relocalization due to oncogene-induced senescence is independent of PML bodies.

**Figure supplement 3**. Validation of RNA-Seq identified expression changes induced by CAPERα and TBX3 KD.

**Figure supplement 4**. IL6 and HDAC9 are direct targets of CAPERα/TBX3.

vivo and that functions as a master regulator of cell proliferation by directly repressing transcription of lncRNA *UCA1*, *p16^{INK}* and other tumor suppressor genes (*Figure 7P*). Overexpression of *UCA1* occurs after loss of TBX3/CAPERα and in OIS (*Figure 7Q*), and is itself sufficient to induce senescence at least in part, by disrupting the interaction of *p16^{INK}* mRNA with hnRNP A1 leading to increased *p16^{INK}* mRNA stability (*Figure 7P,Q*). Disrupting the CAPERα/TBX3 complex by decreasing the amount of either TBX3 or CAPERα, or by CAPERα mislocalization during OIS, coordinately increases activity of multiple pro-senescence targets at both the transcriptional and post-transcriptional levels in a reinforcing mechanism.

Increased CAPERα has been reported in human breast cancers and a shift from cytoplasmic to nuclear localization correlates with transition from pre-malignant to malignant lesions (*Mercier et al., 2009*). In contrast, CAPERα co-activates vRel mediated transcription but inhibits vREL transforming activity in vitro (*Dutta et al., 2008*). It is likely that anti- or pro- oncogenic activity of CAPERα is determined by cell type and the interacting protein(s) present in a given context; our results suggest that CAPERα has oncogenic potential in primary cells since loss of CAPERα/TBX3 induces premature senescence, a vital tumor suppressor mechanism. CAPERα binds to regulatory chromatin domains via TBX3 but dissociates from these domains and becomes concentrated in large intranuclear foci prior to senescence induced by loss of TBX3 or during OIS. Future efforts will define the composition of CAPER + nuclear foci and the role of this nuclear subdomain during senescence induction.

The TBX3 RD is required for TBX3 to interact with CAPERα (this study), immortalize primary fibroblasts and confer senescence bypass (*Carlson et al., 2001*). Since loss of CAPERα activates target gene transcription despite continued TBX3 occupancy, it is the CAPERα/TBX3 complex (interacting via

TBX3 RD) that represses pro-senescence target loci. It will be important to determine if previously identified targets of TBX3 transcriptional repression are actually regulated by this complex.

Additional studies are warranted to determine the precise mechanisms whereby histone status is regulated by CAPERα/TBX3: TBX3 is known to interact directly with HDACs (*Yarosh et al., 2008*), but there are no reports of it or CAPERα interacting with histone methyltransferases or demethylases. Our recently published Mass Spec screen for Tbx3/TBX3 interactors did not identify such factors however, the screen cannot be considered exhaustive as we did not reproducibly detect HDACs or transcription factors previously reported to interact with Tbx3. Future studies to specifically determine whether TBX3 and/or CAPERα interact with, recruit, or modify the function of EZH2, SUV39 and other methyl-transferases will be informative.

Previous studies showed that TBX3 represses transcription of *p14^ARF* (upstream of p53) (*Bamshad et al., 1997*; *Fan et al., 2009*; *Kumar et al., 2014*), yet embryonic lethality and mammary phenotypes of *Tbx3* mutants are p53-independent (*Jerome-Majewska et al., 2005*). Our findings reconcile these observations because CAPERα/TBX3 represses *p16^INK*, the p16/RB pathway is activated in *Tbx3−/−* embryos, and knockdown of either RB or p16 (but not p53) prevents senescence after loss of CAPERα/TBX3. Furthermore, *Tbx3−/−* and *Cdk2−/−;Cdk4−/−* mutant embryos share multiple phenotypes including RB hypo-phosphorylation, reduced E2F-target gene expression, decreased proliferation and premature senescence of MEFs (*Berthet et al., 2012*; *Frank et al., 2012*, *2013*). Our discoveries of multiple CAPERα/TBX3 binding sites across the *CDKN2A* locus, and altered chromatin marks after TBX3 and CAPERα KD, indicate that the complex directly represses transcription by regulating chromatin structure. In total, the data conclusively demonstrate that p16 elevation, *CDK2* and *CDK4* down-regulation, and RB hypophosphorylation mediate senescence downstream of CAPERα/TBX3 loss of function in primary human cells and *Tbx3* null mutant embryos. When combined with the pleiotropic effects of CAPERα/TBX3 on *UCA1, DUSP4, IL6, HDAC9* and other pathways, it is clear why loss of this repressor induces senescence.

TBX3 may function in nuclear organization and structure: severe changes in nuclear morphology and mislocalization of both CAPERα and laminβ1 are apparent in *Tbx3−/−* MEFs after just one passage, prior to other signs of senescence. Progeria is a rare disease in which *LMNA* mutations induce cellular and organismal senescence in part by altering stoichiometry and interactions of type A and B Lamins. Progeria fibroblasts have decreased expression of *TBX3*, TBX3 interacting proteins, and TBX3 targets (*Csoka et al., 2004*). LMNβ1 is a TBX3 interacting protein (*Kumar et al., 2014*) and expression of *LMNA, LMNβ1,* and *LMNβ2* is disrupted by TBX3/CAPERα KD (*Figure 7—source data 1–3* and *Figure 7—figure supplement 3*). TBX3 may regulate *LMN* gene expression and physically interact with Lamins to influence nuclear homeostasis.

There are many downregulated genes common to the senescence responses triggered by RAS^G12V and loss of CAPERα/TBX3 however, upregulated transcripts and pathways are largely distinct (*Figure 7N'*). This is likely attributable to the presence of direct targets of CAPERα/TBX3 repression in the upregulated data set. It will be informative to determine which Jak-STAT, TLR, and TGFβ pathway members (*Figure 7N*) are direct CAPERα/TBX3 targets, as the complex roles of these pathways in the senescence associated secretory phenotype, inducing or enforcing autocrine and paracrine senescence, and tumor progression are emerging (*Hubackova et al., 2010*; *Senturk et al., 2010*; *Hubackova et al., 2012*; *Davalos et al., 2013*).

Recent discoveries of the pervasive functions of lncRNAs as 'signals, decoys, guides and scaffolds' (*Wang and Chang, 2011*), conferred by their ability to interact with other nucleic acids and as protein ligands, has added new layers of complexity to regulation of transcriptional and post-transcriptional gene expression and translation. Although there has been a logarithmic increase in studies exploring lncRNA expression and activity, potential senescence-regulating activities are still largely unexplored. LncRNA *HOTAIR* functions as a scaffold to regulate ubiquitination of Ataxin-1 and Snurportin-1 to prevent premature senescence (*Yoon et al., 2013*). Global alterations in lncRNA expression have been reported in association with replicative senescence (*Abdelmohsen et al., 2013*), and telomere-specific lncRNAs that regulate telomere function during this process have been identified (*Yu et al., 2014*). As this manuscript was in revision, regulation of H4K20 trimethylation of rRNA genes by interaction of quiescence-induced lncRNAs *PAPAS* and Suv4-20h2 was reported (*Bierhoff et al., 2014*). To our knowledge, *UCA1* is the first lncRNA sufficient to induce senescence.

*UCA1* is expressed in bladder transitional cell carcinomas (*Wang et al., 2006*) and influences tumorigenic potential of bladder cancer cell lines (*Wang et al., 2008*; *Yang et al., 2012*). A very recent

study identified hnRNP I as a *UCA1* interacting protein that stabilizes *UCA1* RNA; this interaction was postulated to decrease translation of p27 to support growth of the MCF7 breast cancer line (**Huang et al., 2014**). In contrast, our results support a tumor suppressor/prosenescence function for *UCA1* in primary cells. *UCA1* increases stability of *p16$^{INK}$* mRNA by sequestering hnRNP A1, employing a decoy mechanism that is in some aspects reminiscent of lncRNA *PANDA* sequestering NF-YA transcription factor to prevent activation of proapoptotic p53 targets and promote cell cycle arrest in the DNA damage response (**Wang and Chang, 2011**). In the case of *UCA1* and hnRNP A1 however, the sequestration has a very specific effect: even though *UCA1* expression stabilizes (and hnRNP A1 destabilizes) both *p16$^{INK}$* and *p14$^{ARF}$* mRNAs (**Figure 6A,B**), *UCA1* only disrupts the association of hnRNP A1 with *p16$^{INK}$* mRNA (**Figure 6C** and **Figure 6—figure supplement 4**). In proliferating cells, abundant hnRNP A1 binds with *p16$^{INK}$* mRNA resulting in *p16$^{INK}$* degradation. In senescing cells, *p16$^{INK}$* mRNA levels increase via reinforcing mechanisms of increased transcription and stability: loss of CAPERα/TBX3 activates transcription of *p16$^{INK}$* and *UCA1*, in turn, *UCA1* sequesters hnRNPA1.

We recognize that the systems we employed (primary HFFs, mouse embryos and MEFs), while very informative models, provide limited information directly applicable to aging or tumorigenesis without further experimentation. Our data support an important role for CAPERα/TBX3 in regulation of senescence in developmental contexts and, since the CAPERα/TBX3 complex regulates known critical tumor suppressors and there is an increasing literature supporting roles for both TBX3 and CAPERα in tumor biology, this is another worthy area for future investigation. As noted above, expression of *CDKN2A-p14$^{ARF}$* and *CDKN1A-p21$^{CIP}$* are repressed by TBX2 and TBX3 and this is postulated to confer the ability of overexpressed TBX2 and TBX3 to permit senescence bypass of *Bmi1−/−* and SV40 transformed mouse embryonic fibroblasts, respectively (**Jacobs et al., 2000**; **Brummelkamp et al., 2002**; **Prince et al., 2004**). Numerous overexpression studies have suggested a role for TBX3 in breast cancer ((**Liu et al., 2011**) and references therein) and recent papers have reported the tumorigenic and proinvasive effects of overexpressed TBX3 in melanoma cells (**Peres et al., 2010**; **Peres and Prince, 2013**) which may derive in part from TBX3 repression of E-cadherin expression (**Rodriguez et al., 2008**). More relevant to our work on the importance of the CAPERα/TBX3 complex to prevent senescence and regulate cell proliferation are reports that Tbx3 improves the pluripotency of iPS cells (**Han et al., 2010**) and prevents differentiation of mouse ES cells (**Ivanova et al., 2006**).

In conclusion, CAPERα/TBX3 acts as a master regulator of cell growth and fate, exerting pleotropic effects by at least two modes of action: (1) regulating chromatin structure and transcription of both coding and non-coding genes and, (2) modulating mRNA stability by altering the association of RNA binding proteins with target transcripts via *UCA1*. Further exploration will identify tissue-specific *UCA1* targets and binding proteins, and determine whether the ability of TBX3 to confer senescence bypass in other contexts requires CAPERα interaction and/or *UCA1* repression. Mining the pathways regulated by *UCA1* and CAPERα/TBX3 will reveal factors that control cell proliferation and fate during development and disease and thus constitute novel cancer therapeutic targets.

## Material and methods

### Mass spectroscopy

Mass spectroscopy as in **Kumar et al., (2014)**

### Protein extraction and immunoprecipitation

Dignam lysates were prepared and incubated for 4 hr at 4°C with the appropriate antibody followed by 2 hr at 4°C with the pre equilibrated Dynabeads Protein G (Invitrogen). Immune complexes were collected and washed three times with lysis buffer. Pelleted beads were resuspended in 6X Laemmli buffer and subjected to SDS-PAGE analysis followed by immunoblotting with specific antibodies.

Input lanes contain 5% of protein lysate used for IP; the rest was used in the IP and of the IP'd material, 25% was loaded onto the gel for immunoblotting.

### Antibodies

Tbx3 (**Frank et al., 2012**, **2013**), TBX3 (SC-17871,MAB10089,A303-098A), CAPERα (A300-291A), GST (SC-33613), LaminB1 (SC-56144), C-Myc (SC-40), R-IgG (SC-2027), m-IgG (SC-2025), Anti-Flag (Sigma, F3165), H3K9me3 (Cell Signaling, 9754), H3K4me3 (Cell Signaling, 9751), H3K27me3 (Cell Signaling, 9733), H3K9ace (Cell Signaling, 9649), H4K5ace (Cell Signaling, 9672), H3K14ace (Cell Signaling, 4353), p-RB -Ser 810--811 (SC-16670), p-RB -Ser 795 (SC-7986), p-RB -Ser 780 (SC-12901), Rb1

(SC-73598), H3S10P (SC-8656), H2A K119ub (8240S), p21 (SC-756), p53 (Invitrogen 134100), Cyclin D1 (SC-753), Cyclin D2 (SC-754), Cyclin D3 (SC-755), Cyclin E (SC-20648), CDK2 (SC-6248), CDK4 (SC-601) CDK6 (SC-177), hnRNP K (SC-53620), hnRNP C1/C2 (SC-32308), hnRNP H (SC-10042), hnRNP U (SC-32315), hnRNP A2/B1 (SC-53531), hn RNP A1 (SC-32301), and hnRNP D1 (AB-61193).

## MBP pull down assay

Amylose bound MBP and MBP-tagged TBX3 affinity columns were prepared as per the procedure (E8022S, NEB) described in the manufacturer's protocol. These beads were incubated with 5 and 10 μg of GST and GST-CAPER at 4°C for 8 hr. Bound proteins were eluted with reduced glutathione and analyzed by Western blotting with anti-CAPER antibody.

## Cell transfection

Transfections were performed in HEK293 or EBNA-293 cells with Lipofectamine 2000 (Invitrogen) or in Human fibroblasts with X-tremeGENE HP DNA transfection Reagent (Roche) as per the manufacturer's recommendations.

## Plasmids

Wild-type Tbx3 and exon 7 missense, deleted repressor domain (Tbx3ΔRD1), and Tbx3ΔNLS were generated by PCR amplification and cloned into pcDNA3.1. C-terminal deletion constructs Tbx3 1-655, Tbx3 1-623, Tbx3 1-565, Tbx3 1-470 were generated by PCR amplification and cloned into pCS2 with an N-terminal Myc tag. Tbx3 L143P and N277D point mutants were kind gifts of Phil Barnett. *UCA1* and CAPERα cDNAs were cloned into pCDN3.1 and PQCXIH for over- expression studies, respectively. Sequence of all plasmids was confirmed. Tbx3 L143P and N277D point mutants plasmids were kind gifts of Phil Barnett. Wild-type CAPERα was generated by PCR amplification and then cloned into pQCXIH retroviral vector; sequence was confirmed. Full length *UCA1* was amplified by PCR and then cloned into pcDNA3.1 vector; sequence was confirmed.

> UCA1 Cloning FP: AGTTGCGGCCGCTGACATTCTTCTGGACAATGAG
> UCA1 Cloning RP: TCCTGCGGCCGCTTGGCATATTAGCTTTAATGTAG
> CAPERα Cloning FP: CATCGCGGCCGCATGGCAGACGATATTGATATTG
> CAPERα Cloning RP: ACGTGGATCCTCATCGTCTACTTGGAACCAGTAG

## Immunofluroscence

E10.5 embryos were harvested in PBS followed by overnight fixation at 4°C in 4% paraformaldehyde and processed for 7 μm cryosections. For cell lines, human fibroblasts were cultured on 8-well chamber slides (BD Flacon) and processed for Immunohistochemistry. Immunohistochemistry was performed using primary antibodies listed above and detected using donkey anti-goat or anti-rabbit Alexa fluor 594 (1:500) and goat anti-mouse Alexa fluor 488(1:500) from Invitrogen. Nuclei were stained with Hoechst or DAPI. Slides were imaged with a Nikon ARI inverted confocal microscope at the University of Utah Imaging Core.

## Retroviral transduction and selection of stable cells

shRNA oligonucleotides (see sequences below) were annealed and cloned into the pGFP-B-RS, pRFP-C-RS (Origen) vector and PMK0.1 vector. shRNA against luciferase served as a negative control. High-titer retrovirus was produced by transfection of shRNA retroviral construct along with gag/pol and VSVG encoding plasmids into EBNA-293 cells by lipofectamine 2000 reagent as per the manufacturer's protocol. Virus containing supernatant was collected after 48 hr of transfection and filtered through 0.45-μM filters (Fisher 09-720-4). HEK293 or HFFs were incubated with DMEM containing polybrene (8 mM) and 500 μl of TBX3 or CAPERα shRNA encoding retrovirus. 24 hr post infection, cells were split to lower densities and blasticidin or puromycin antibiotic selection applied for 2 days. Stably integrated colonies were selected and analyzed for knock down efficiency by western analysis using Tbx3 or CAPERα antibody.

> TBX3 shRNA A: targets *TBX3* exon 7
> TBX3 shA FP: CCGG GACCATGGAGCCCGAAGAA ttcaagaga TTCTTCGGGCTCCATGGTC TTTTTG
> TBX3 shA RP: AATTCAAAAA GACCATGGAGCCCGAAGAA tctcttgaa TTCTTCGGGCTCCATGGTC
> TBX3 shRNA B: targets *TBX3* exon 5
> TBX3 shB FP: CCGG CAGCTCACCCTGCAGTCCA ttcaagaga TGGACTGCAGGGTGAGCTG TTTTTG
> TBX3 shB RP: AATTCAAAAA CAGCTCACCCTGCAGTCCA tctcttgaa TGGACTGCAGGGTGAGCTG

CAPERα shRNA A: targets *CAPERα* (gene name *RBM39*) exon 5
CAPERα shA FP: CCGG GACAGAAATTCAAGACGTTttcaagagaAACGTCTTGAATTTCTGTCTTTTTG
CAPER shA RP: AATTCAAAAA GACAGAAATTCAAGACGTT tctcttgaa AACGTCTTGAATTTCTGTC
CAPERα shRNA B: targets *CAPERα* exon 1
CAPER shB P:CCGG AAAGCAAGAGCAGAAGTCGTAttcaagagaTACGACTTCTGCTCTTGCTTTTTTTTG
CAPER shB RP: AATTCAAAAA AAAGCAAGAGCAGAAGTCGTA tctcttgaa TACGACTTCTGCTCTTGCTTT

The pMKo.1 puro RB and pMKo.1 puro p53 shRNA vectors were a kind gift of William Hahn obtained via Addgene.

pRB shRNA: Addgene #10670
p53 shRNA: Addgene #10672
p16 shRNA: Addgene #22271

Efficacy and specificity of the pRb, p53, and p16 shRNAs was validated with second shRNAs, and these reagents have been used extensively by many investigators in the years since their initial publication (*Masutomi et al., 2003*; *Stewart et al., 2003*; *Boehm et al., 2005*; *Haga et al., 2007*; *Hong et al., 2009*; *Elzi et al., 2012*).

*UCA1* shRNA: targets *UCA1* exon 3
UCA1 shA FP:
GATCCGTTAATCCAGGAGACAAAGAtcaagagTCTTTGTCTCCTGGATTAACTTTTTTGGA
*UCA1* shA RP:
AGCTTCCAAAAAAGTTAATCCAGGAGACAAAGActcttgaTCTTTGTCTCCTGGATTAACG
Senescence associated β-galactosidase assay
Performed as per the manufacturer's protocol (9860, Cell Signaling).
Population doubling assay/3T5 growth curves (*Figure 2E,F,R*)

Primary HFFs were plated in a 10-cm dish and transduced with retrovirus. After 24 hr, cells were cultured with antibiotic selection (puromycin or blasticidin) for an additional 24–72 hr. On day 0 of the 3T5 growth curve, cells were trypsinized, counted and 500,000 cells were then plated per 10-cm dish. This procedure was repeated every 3 days for 15 days. Population doublings were calculated by (logN1/log2) − (logN0/log2) N1 = current cell count, N0 = Initial cell count. Curves shown in *Figure 2* are representative of two independent experiments.

## Cell count (*Figure 5C*)
Primary HFFs were plated in 6-well dishes and transfected at 70% confluence. At days noted in the figure, cells were trypsinized and counted using a hemocytometer.

## Crystal violet assay/optical density method of cell quantitation
$5 \times 10^5$ cells were plated per well in 6-well tissue culture plates. At times indicated, medium was removed and cells were washed with PBS, and fixed for 10 min in 10% formalin solution. Cells were rinsed 5X with distilled water, and then stained with 100 µl 0.1% crystal violet solution for 30 min, rinsed 5X in water and dried. Cell-associated crystal violet dye was extracted with 500 µl of 10% acetic acid. Aliquots were collected and optical density at 590 nm measured. Each point on the curve shown represents three independent plates.

## Senescence marker gene expression in TBX3 and CAPERα KD fibroblasts
Primary HFFs were incubated with TBX3 or CAPERα or Control shRNA encoding retrovirus medium with fresh virus added every 8 hr for 48 hr, followed by antibiotic selection for 6 days. 6 days after selection, floating cells were discarded and adherent cells were utilized for senescence associated β-gal assay or preparation of RNA.

## RNA isolation and reverse transcription-PCR analysis
Total RNA was prepared using the RNeasy RNA isolation kit (Qiagen) or NucleoSpin RNA II Kit (Clontech) and cDNA was synthesized by cDNA EcoDry Premix Double Primed (Clontech) kit. Q-RT-PCR was performed with SoFast Evagreen Supermix (Bio-Rad) as per manufacturer's protocol.

## RT-PCR primer sequences

TBX3: TGAGGCCTTTGAAGACCATG, TCAGCAGCTATAATGTCCATC
CAPERα: CGGAACAGGCGTTTAGAGAA, TGGCACTGCTCAACTTGTTC
CDK2: GCTTTCTGCCATTCTCATCG, GTCCCCAGAGTCCGAAAGAT
CDK4: ACGGGTGTAAGTGCCATCTG, TGGTGTCGGTGCCTATGGGA
P21: TCAGAGGAGGCGCCATGT, TGTCCACTGGGCCGAAGA
CDC2: GGGGATTCAGAAATTGATCA, TGTCAGAAAGCTACATCTTC
MDM2: ACCTCACAGATTCCAGCTTCG, TTTCATAGTATAAGTGTCTTTTT
MAPK14: TTCTGTTGATCCCACTTCACTGT, ACACACATGCACACACACTAAC
CDKN2C: CAATGGCTCAGTTTTGCTGAATAA, GTAAGATCTGCCTGCCAAAAGC
CDKN2B: AACGGAGTCAACCGTTTCGG, TGTGCGCAGGTACCCTGCA
P16: CAACGCACCGAATAGTTACG, AGCACCACCAGCGTGTC
SerpinE1:CCGGAACAGCCTGAAGAAGTG, GTGTTTCAGCAGGTGGCGC
P14ARF: CCCTCGTGCTGATGCTACTG, ACCTGGTCTTCTAGGAAGCGG
MCM3: CCTTTCCCTCCAGCTCTGTC, CTCCTGGATGGTGATGGTCT
TGFb: AAGGACCTCGGCTGGAAGTG, CCCGGGTTATGCTGGTTGTA
EGR1: CCAGGAGCGATGAACGCAAGCGGCATACCAAG, GGAGTACGTGGTGGCCACCGACGGGGACCC
E2F1: ATGTTTTCCTGTGCCCTGAG, ATCTGTGGTGAGGGATGAGG
E2F2: GGCCAAGAACAACATCCAGT, TGTCCTCAGTCAGGTGCTTG
IL6R: CATTGCCATTGTTCTGAGGTTC, AGTAGTCTGTATTGCTGATGTC
GSK3b: ACTCCACCGGAGGCAATTG, GCACAAGCTTCCAGTGGTGTT
UCA1:GAAATGGACAACAGTACACGCATATGGGGC, CCTGTTGCTAAGCCGATGATACATTACCCT
HPRT: GCTGGTGAAAAGGACCTCT, CACAGGACTAGAACACCTGC
PCNA: AAGAGAGTGGAGTGGCTTTTG, TGTCGATAAAGAGGAGGAAGC
CHK2: CTTATGTGGAACCCCCACCTAC, CAGCACGGTTATACCCAGCA
PMAIP1: GTTTTTGCCGAAGATTACCG, CAATGTGCTGAGTTGGCACT
MYC: CTCCCTCCACTCGGAAGGA, GCATTTTCGGTTGTTGCTGAT
CDKN2D: CAACCGCTTCGGCAAGAC, CAGGGTGTCCAGGAATCCA
P53: CCTCACCATCATCACACTGG, TCTGAGTCAGGCCCTTCTGT
RB: TGTGAACATCGAATCATGGAA, TCAGTTGGTGGTTCTCGGTC
CXCL10: GAAATTATTCCTGCAAGCCAATTT, TCACCCTTCTTTTTCATGTAGCA
IFNB1: GAATGGGAGGCTTGAATACTGCCT, TAGCAAAGATGTTCTGGAGCATCTC
ATF3: GTTTGAGGATTTTGCTAACCTGAC, AGCTGCAATCTTATTTCTTTCTCGT
DUSP2: GGCCTTTGACTTCGTTAAGC, CCACCTCAGTGACACAGCAC
CREB5: CGTGCCTCCTTGAAACAAGCCATT, ATGAAACACCAGCACCTGCCTAGA
HDAC9: AGTGTGAGACGCAGACGCTTAG, TTTGCTGTCGCATTTGTTCTTT
SP140: TGGGTCAGTTTCTTGTTTATCTGC, AGCAGGCTAGAAGCAAGCTC
EGR2: TTGGTGCCTTGTGTGATGTAGAC, CTTTCCATAAGGCAACCCATTT
HMGA2: GTCCCTCTAAAGCAGCTCAAAA, CTCCCTTCAAAAGATCCAACTG
BIRC5: CATGGTAGGTGCAGGTGATG, CATGGTAGGTGCAGGTGATG
ASF1: GGTTCGAGATCAGCTTCGAG, CATGGTAGGTGCAGGTGATG
WDR66: CCGAGAAGCAACAGGAGAAA, CTGTGTCTCCAAACGGATCA
CDC25C: GACACCCAGAAGAGAATAATCATC, CGACACCTCAGCAACTCAG
CENPF: CGAAGAACAACCATGGCAACTCG, TTCTCGGAGGATGGTGCCTGAAT
LAMA2: AATTTACCTCCGCTCGCTAT, CCTCCAATGTACTTTCCACG
LMNB1: AAGCAGCTGGAGTGGTTGTT, TTGGATGCTCTTGGGGTTC
LMNB2: GCTCTGACCAGAACGACAAGG, CCAGCATCTTCCGGAACTTG
CDC20: TCCAAGGTTCAGACCACTCC, GATCCAGGCCACAGACCATA
DUSP5: GCTCGCTCAACGTCAACCTCAACTCGGTG, AGTGGCGGCTGCCCTGGTCCAGCACCACC
DUSP4: CCTGGCAGCCATCCCACCCCCGGTTCCCC, GCTGATGCCCAGGGCGTCCAGCATGTCTCTC
mTbx3: TGAGGCCTCTGAAGACCATG, TCAGCAGCTATAATGTCCATC
mSerpinE1: AGCCAACAAGAGCCAATCAC, GGATTCTCGGAGGGGTAAAG
mIL6: GATGGATGCTACCAAACTGGA, CCAGGTAGCTATGGTACTCCAGAA
mP21: TCCACAGCGATATCCAGACA, GGCACACTTTGCTCCTGTG

mCdc2: CTGCAATTCGGGAAATCTCT, TCCATGGACAGGAACTCAAA
mReprimo: CTTACGGACCTGGGACTTTG, CCAGCACTGAATTCATCACG

## MEF isolation from WT and *Tbx3* null embryos

All steps were performed under aseptic conditions. Pregnant female mice were euthanized and 13.5-day-old embryos were isolated from the uterus. Embryos were washed in sterile PBS in 60-mm tissue culture dish at room temperature and transferred into 15-ml sterile falcon tube containing 1 ml of 50% trypsin in DMEM medium. Embryos were minced using fine scissors followed by gentle pipetting with 1 ml pipette tips and dispersed into cell suspensions in 5 min. Suspensions were plated into 10-cm plates in 10 ml of DMEM with 5% FBS and penicillin/streptomycin and incubated for 8 hr in $CO_2$ incubator. Culture medium was replaced with fresh medium every day for 3 days. Passage 0 refers to the stage when cell suspension from the embryos was put into cell culture and subsequent passages are numbered.

## Chromatin immunoprecipitation (ChIP)

Performed as per the manufacturer's protocol (9003S, Cell Signaling).

### ChIP primers

UCA1 FP1: GGCTCTCGAGTCAAGATAATTCACTTAC
UCA1 RP1: GGCACATCTTTGTTGTCTGAAAGGGAT
UCA1 FP2: CACCTCTTTCTTGCCTCCTTGGATATATT
UCA1 RP2: CACTTACTTACTTATAATAGAGTCAGGGTCT
UCA1 FP3: CCAGGAGCTGATATTCATGACCCTCCA
UCA1 RP3: CTTGGCTCCTGTAGGCCACCTGGACAT
DUSP4 FP: CGAGGGCACCGGTACCCGCCGGGTCTCTCC
DUSP4 RP: GGACTAGGGTGAGCACAAGCCTTGAGCGC
P16 1A FP: CGACCGTAACTATTCGGTGCGTTGGGCAGC
P16 1A RP: GCTCTGGCGAGGGCTGCTTCCGGCTGGTGC
P16 2A FP: GAGCAGGACGCGGTGGCTCACACCTGTAAT
P16 2A RP: CAGGCATGCGCCACCAAGCCCCGCTAATT
P16 3A FP: CCTCGGGGTACCTCTCAATTAGCTGTGTA
P16 3A RP: AGTTCGAGACAAGCCTAGCCAACATAGTG
P16 4A FP: GAAACTCTACCATGGATTCCTACATCAAG
P16 4A RP: GCACAATGTGCAGGTTTGTTACATATGTAT
P16 5A FP: CCAGTCTCAGATTTCCTATGTGCAAAATG
P16 5A RP: GGTTTGAACCCTGGCAGTCTGACTGTAG
P16 6A FP: GCGGTGGTTATAGATTTTGTCACAAGAG
P16 6A RP: ACTCTGGAACACTACCTTCTCAAGTATC
P16 7A FP: ACCCCGATTCAATTTGGCAG
P16 7A RP: AAAAAGAAATCCGCCCCCG
P14ARF: FP: GCCGAATCCGGAGGGTCACCAAGAACCTGC
P14ARF: RP: GTGCGCAGGGCTCAGAGCCGTTCCGAGATCT
CDK2 FP: GATGGAACGCAGTATACCTCTC
CDK2 RP: AAAGCAGGTACTTGGGAAGAGTG
CDK4 FP: GTGGACCGAAAAGGTGACAGGATC
CDK4 RP: GGGCGGGGCGAACGCCGGACGTTC
P21 −324 to −676 FP: CCCGGAAGCATGTGACAATC
P21 −324 to −676 RP: CAGCACTGTTAGAATGAGCC
P21 −677 to −981 FP: GGAGGCAAAAGTCCTGTGTTC
P21 −677 to −981 RP: GGAAGGAGGGAATTGGAGAG
P21 −964 to −1340 FP: CTGAGCAGCCTGAGATGTCAG
P21 −964 to −1340 RP: CACAGGACTTTTGCCTCCTG
P21 −1335 to −1688 FP: GAAATGCCTGAAAGCAGAGG
P21 −1335 to −1688 RP: GCTCAGAGTCTGGAAATCTC
CDKN1B FP: CGGCCGTTTGGCTAGTTTGTTTGT
CDKN1B RP: GGAGGCTGACGAAGAAGAAGATGA

HDAC9CHIPFP: GGCTCAGGCCGACCATTGTTCTATTTCTGT
HDAC9CHIPRP: CCTGAGGAGAAGCAGCAGAGGATCAAC
IL6CHIPFP: GAACCAAGTGGGCTTCAGTAATTTCAGG
IL6CHIPRP: CATCTGAGTTCTTCTGTGTTCTGGCTCTC
P14ARF FP: CCCTCGTGCTGATGCTACTG
P14ARF RP: ACCTGGTCTTCTAGGAAGCGG
TGFB1 FP: GATGGCACAGTGGTCAAGAGC
TGFB1 RP: GAAGGATGGAAGGGTCAGGAG
RB FP: GGCGGAAGTGACGTTTTC
RB RP: CCGACTCCCGTTACAAAAAT
MYC FP: AAGATCCTCTCTCGCTAATCTCC
MYC RP: AGAAGCCCTGCCCTTCTC
E2F1 FP: GGCTACAGGTGAGGGTCACG
E2F1 RP: GAGCGCCGCCACAATTGGCT
CDKN2D FP: TCCCTTTCTTCACGGTGCTT
CDKN2D RP: GCGTCGCTCCTGATTGGTC
CDK2 FP : AAGCAGGTACTTGGGAAGAGTGTTCAGC
CDK2 RP: CAACTTGAAACAATGTTGCCGCCTCC
MDM2 FP: GGCCTACCCAAAGTGATGGGATTACAAG
MDM2 RP: GCCGCTGGAGTTGTACCCAAATGAGTTA

## siRNA knockdown

For differential display (*Figure 4*), HEK293 cells were transfected with control siRNAs (Sense; 5'-CAGCGACUAAACACAUCA-3' Antisense; 5'-UUGAUGUGUUUAGUCGCUGTT-3') or TBX3 specific siRNA A (Sense: GACCAUGGAGCCCGAAGAA, Antisense: UUCUUCGGGCUCCAUGGU) or CAPERα-specific siRNA (Sense: GACAGAAAUUCAAGACGUU, Antisense: AACGUCUUGAAUUUCUGUC) using lipofectamine 2000 (Invitrogen) or X-treme GENE HP DNA transfection reagent as per manufacturer's instructions.

HNRNP A1 siRNA for knockdown in HFFs (*Figure 6*) was obtained from Cell Signaling (cat. #7668).

## Oncogene-induced senescence with constituitively active RAS

$^{V12G}$RAS virus was produced with pBABE-$^{V12G}$RAS as per the procedure described above. HFFs were transduced with RAS virus and incubated with antibiotic selection medium (puromycin 2 µg/ml) for 4–5 days.

## RNA immunoprecipitation (RIP) and RIP-PCR

For RNA immunoprecipitation, 10 million cells were lysed in 1 ml of NP-40 lysis buffer (50 mM Tris HCl, ph7.4, 150 mM NaCl, 1% NP-40 and Protease inhibitor cocktail). Lysate was cleared by centrifugation at 12,000 RCF for 15 min. Cleared lysate was immunoprecipitated independently with 5 µg of anti-hnRNP A1, anti-hnRNP D, Anti-hnRNP A2/B1, Anti-hnRNP C1/C2, Anti-hnRNP K, mIgG and R-IgG antibodies. Immune complexes were incubated with 30 µl of pre-equilibrated Dynabeads for 4 hr at 4°C. Dynabead purified immune complexes were subjected to Proteinase K digestion at 37°C for 1 hr followed by NucleoSpin RNA II purification kit and cDNA was prepared by RNA-to-cDNA EcoDry Premix kit (Clontech). cDNA was used as a template in PCR amplifications with gene specific primers.

## mRNA stability assays

TBX3, CAPERα, or Control shRNA KD, PS and RAS HFFs were cultured in 6-well culture dishes for 2 days to 80% confluence. Then Actinomycin D was added to a final concentration of 5 mg/ml to suppress transcription. At 0, 1, 2, and 4 hr after addition of Actinomycin D, equal numbers of cells were harvested from each sample and mRNA was prepared by nucleoSpin RNA II purification kit and cDNA was prepared by RNA-to-cDNA EcoDry Premix kit (Clontech) followed by qRT-PCR for specific transcripts.

## P16$^{INK}$ mRNA northern blot

HFFs were transfected with pcDNA3.1 control or *UCA1* expression plasmids as described above, incubated +/− Actinomycin D, and total cellular RNA was harvested at 0, 1, 2, and 4 hr post treatment. For

northern blot analysis, 5 µg total RNA from each time point was electrophoresed through a 1% agarose gel. The RNA was blotted onto Hybond-N+ membrane (Amersham Pharmacia), and membranes were UV crosslinked. Membranes were hybridized for 18 hr with (*Torres et al., 2003*) P-labeled probes. Probes were generated by end-labeling DNA oligonucleotides containing following sequences complementary to *p16^INK* mRNA:

1. 5′ GAGGAGGTGCTATTAACTCCGAGCATTAGCGAATGTGGC
2. 5′ AATCCTCTGGAGGGACCGCGGTATCTTTCCAGGCAAGGGG
3. 5′AAGGCTCCATGCTGCTCCCCGCCGCCGGCTCCATGCTGCT

End-labeling reactions were performed using T4 polynucleotide Kinase (NEB) according to the manufacturer's directions. The hybridized blots were washed, and autoradiographs were developed as per standard procedure. Band intensities were measured by Image J analysis, and densitometric vales were plotted as bar graphs.

## RNA-Seq analysis of TBX3 and CAPERα KD HFFs

HFFs were incubated with TBX3 or CAPER α shRNA encoding retrovirus for 48 hr followed by incubation for an additional 48 hr in selection medium. Total RNA was isolated and purity was assessed. Poly-A RNA was purified, fragmented, primed with random hexamers and used to generate first strand cDNA using reverse transcriptase. Samples that passed quality control steps were used for Illumina library preparation using the Illumina TruSeq RNA Sample Prep protocol. All libraries were sequenced (with barcoding) on a single lane of an Illumina HiSeq instrument for 50 cycles from a single end. A total of 177,155,781 reads were produced in total for all 10 libraries (median 17,348,374 reads). Base calling was performed using Illumina software.

## Bioinformatics analysis

Sequence reads were aligned (98.5% mapped) to the human genome build 37.2 with Tophat (v2.0.8b) using default parameters. Aligned reads were assembled into transcripts and their relative abundance was measured using Cufflinks (v2.1.1) with fragment bias correction (frag-bias-correct) and multi-read correction (multi-read-correct). Cufflinks transcript assemblies were based on transcripts of NCBI Homo sapiens annotation release 104 and miRBase release 19 as provided in the Illumina iGenomes data set. Cuffdiff was used to test for differential expression between samples and controls and expression differences were taken as significant if the FDR adjusted p-value was less than 0.05 (Source Data Files 1 and 2). Statistically overrepresented gene ontology/biologic process categories and KEGG pathways were determined using DAVID (*Huang da et al., 2009a*, *2009b*). The hypergeometric test, as implemented in the R statistical language (phyper), was used to test significance of the number of genes found to be co-regulated between samples (*Figure 7—source data 3*).

## Acknowledgements

We thank J Michael Dean and Thomas Vondriska for their support of Drs Kumar and Franklin, respectively. We thank Nikos Tapinos, David Carey, Alana Welm, Kirk Thomas, and Ashley Firment for critical reading of the manuscript. Phillip Barnett kindly provided the Tbx3 DBD mutant plasmids.

## Additional information

### Funding

No external funding was received.

### Author contributions

PKP, AMM, Conception and design, Acquisition of data, Analysis and interpretation of data, Drafting or revising the article, Contributed unpublished essential data or reagents; UE, Conception and design, Acquisition of data, Contributed unpublished essential data or reagents; RS, Acquisition of data, Analysis and interpretation of data, Contributed unpublished essential data or reagents; SF, BM, Conception and design, Acquisition of data, Analysis and interpretation of data; MY, Bioinformatics, Analysis and interpretation of data, Contributed unpublished essential data or reagents; SLL, Training, Acquisition of data, Analysis and interpretation of data

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
