## [Decision Letter]

Thank you for sending your work entitled “Coordinated control of senescence by lncRNA *UCA1* and a novel CAPERα/TBX3 co-repressor” for consideration at *eLife*. Your article has been favorably evaluated by a Senior editor and 2 reviewers, one of whom is a member of our Board of Reviewing Editors.

The Reviewing editor and the other reviewer discussed their comments before we reached this decision, and the Reviewing editor has assembled the following comments to help you prepare a revised submission.

In this study, Kumar et al. report on the role of corepressor complex composed of Caper and TBX3 in control of senescence. They document the existence of a Caper-TBX3 complex and then show that it is required to prevent senescence. They go on to demonstrate that the Caper-Tbx3 functions to prevent senescence by repression of p16 and RB family members. Most interestingly, they show that Caper-Tbx3 targets and repress a long non-coding RNA, UCA1, and that over-expression of UCA1 is sufficient to induce senescence. They provide evidence that p16 mRNA is bound and destabilized by hnRNP A1 and that UCA1 stabilizes p16 mRNA by sequestering hnRNP A1. Activated Ras, which induces senescence in primary cells, disrupts and inactivates the Caper-TBX3 corepressor resulting in UCA1 expression and senescence induction.

In general, the major conclusions of the work are interesting and supported by the presented results. However, there are number of areas in which the manuscript can be improved.

The following issues must be addressed for publication in *eLife*.

1) It is relatively standard in shRNA experiments to validate the results using a second shRNA against the same target gene to rule out off-target effects. Ideally, two shRNAs should be used for all experiments but minimally this should be done for the initial and key experiments. Two shRNAs to Caper and Tbx3 are mentioned in the methods section but it is not clear which were used in the various experiments and whether any experiment was performed with two shRNAs. This same concern applies to the other shRNAs used in the study, such as that directed against p53.

2) In the knockdown/ectopic expression rescue experiments, it is not clear whether the ectopically expressed mRNA is subject to RNAi-mediated degradation. This can affect the interpretation of the results.

For example, in the experiments performed to determine the domain of TBX3 required for interaction with CAPERalpha. Use of shRNAs against TBX3 makes the interpretation of the results difficult, because it might also target some of the TBX3 deletion constructs. It would be preferable to perform these experiments using epitope-tagged TBX3 in the absence of shRNAs against TBX3.

Also, the rescue experiments using ectopic expression of CAPERalpha are difficult to interpret without knowing the region of CAPERalpha mRNA that is being targeted by the CAPERalpha shRNA. Please provide this information for all the shRNAs in the experimental methodology section. As in point 1 above, the authors should also include a second shRNA for their experiments to rule out shRNA-related “off-target” effects.

3) Statistical analyses are missing throughout the manuscript. Because of this it is not clear if some of the results presented in the manuscript are statistically significant. Please provide p-values for all of the results.

4) The reviewers felt that the writing and presentation of the manuscript can be considerably improved. For example, there are uncommon usage of phrases (for example, embryonic mice, rather than mouse embryo; oncogene-induced stress physically dissociate, rather than disrupt; many others).

In general the figure legends are too sparse for the interested reader to fully understand how the experiments were carried out and presented. *eLife* does not have arbitrary word limits so there is no reason that additional details can't be provided. For example, in the mass spec experiment of Figure 1, it is not clear which peaks are those of Caper and there is no information in the legends. In Figure 1 and elsewhere there is no description as to what the black and red arrows indicate.

Finally, in the Method section, I could find no description of the methods used for the RIP experiments.

The following issues would be desirable to address for publication in *eLife*. We encourage the authors to address as many of these as possible.

1) The experiments in Figure 1 E-G seem very tangential to the main conclusions of this study. The authors should consider removing these to help focus the manuscript.

2) For all of the mRNA stability experiments can the authors provide a half-life estimate for the mRNA and discuss whether the half-life measurement is consistent with those obtained in other studies. Also, it would be desirable to confirm at least the key half-life experiments, such as those measuring p16 mRNA, by northern blot to rule out possible amplification artifacts that can arise by PCR.

3) Figure 6. Can the increased hnRNPA1 binding to UAC1 following Ras expression be attributed to increased UAC1 RNA levels?

4) Why is H3K27me3 levels measured in some but not all experiments?

5) Figure 7. In contrast to what is stated in the text, it is not clear that Caper levels markedly decrease in the cytoplasm. Also in panels J and L the magnifications are not equivalent.

6) The authors use differential display to identify UAC1. This is an older and currently infrequently used procedure. Can the authors provide a rationale for their use of differential display?

7) The authors show that CAPERalpha/TBX3 regulates histone modifications associated with p16 and UCA1 promoters. However, it is not clear how CAPERalpha/TBX3 achieves this. Are the changes in histone modifications a direct consequence of CAPERalpha/TBX3 activity? It would be desirable if the authors had results that address this issue but minimally the various possibilities should be discussed. Additionally, it would be desirable for these experiments if total RNA polymerase II and elongating RNA polymerase II ChIP results could be included.

8) The results describing the role of CAPERalpha/TBX3 in premature senescence and oncogene-induced senescence are interesting but do not provide any insight into the role of this CAPERalpha/TBX3 pathway in either the process of aging or RAS-driven tumor initiation. Is there a tumor type where the authors can show this regulation occurs and modulates RAS-induced tumor initiation? This would add an important physiological context. Fibroblast-based experiments do not provide meaningful advances in our understanding for the role of CAPERalpha/TBX3 mediated senescence repression to either aging or tumorigenesis. Also, it will be desirable to determine the state of telomerase expression/activity, telomere length and state of shelterin proteins in the experiments where premature senescence is measured.

9) There are several previous reports that TBX3 represses cellular senescence. These previous should be discussed. Similarly, there are several reports on the role of long non-coding RNAs in the regulation of cellular senescence and these studies should also be discussed.

---

## [Author Response]

1*) It is relatively standard in shRNA experiments to validate the results using a second shRNA against the same target gene to rule out off-target effects. Ideally, two shRNAs should be used for all experiments but minimally this should be done for the initial and key experiments. Two shRNAs to Caper and Tbx3 are mentioned in the methods section but it is not clear which were used in the various experiments and whether any experiment was performed with two shRNAs. This same concern applies to the other shRNAs used in the study, such as that directed against p53*.

For clarity, we have now separated the data on effectiveness and specificity of the CAPERα and TBX3 shRNAs (two different shRNAs for each) and they are now presented in Figure 2—figure supplement 1 and Figure 2—figure supplement 2, respectively. The two different shRNAs have comparable effects on SAβgal activity; in both cases, subsequent experiments were performed with TBX3 shRNA A and CAPERα shRNA A and information as to their target sites within the mRNAs is now provided in the Methods.

The p53, p16 and pRb shRNAs we used were obtained from Addgene (please see Methods for specific information); their efficacy and specificity has been validated against second shRNAs and these reagents have been used extensively by many investigators in the years since their initial publication including (but not limited to) [1-5]. We have now cited the original use of these reagents in the Results section and the Methods of the revised manuscript.

*2) In the knockdown/ectopic expression rescue experiments, it is not clear whether the ectopically expressed mRNA is subject to RNAi-mediated degradation. This can affect the interpretation of the results*.

We completely agree that this is an important consideration and apologize that our strategies to deal with this issue were not made clear.

In our manuscript that was in press at the time of the original submission and is now published [6], we show that all the mutant Tbx3 (mouse) proteins are produced in TBX3 knockdown HEK293 cells (Figure 2 of the published manuscript). We apologize that we did not provide access to this data to the reviewers of the present manuscript and have remedied this by citing the paper, extensively revising the text, legend, and figure in the revised manuscript to more clearly explain the experimental design.

The DNA binding domain mutants and the ΔRD and exon7 missense mutations are untagged while the C-terminal deletion mutants are Myc-tagged. In order to assay the interactions of the untagged proteins with CAPERα, we needed to use TBX3 shRNA knockdown HEK293 cells. Figure 1, panel J shows that CAPERαis present and can be IP’d despite knockdown of endogenous human TBX3 and subsequent overexpression of mutant mouse Tbx3 proteins. Panel J’ shows that the Tbx3 point mutant proteins (L143P, N227D) coIP with CAPERα in an anti-CAPERα IP. We did not think it necessary to show that the mutant proteins were expressed in the knockdown cells since they are detected in the IP with CAPERαand the data demonstrating their production in the knockdown cells are published.

The deletion constructs were Myc- tagged and did not require assay in knockdown cells and so panels K/K’ are in wild type HEK293 cells. Thus, failure to detect interaction is not attributable to knockdown of the overexpression mutant mRNAs. In this case, because the IP was performed for Myc-Tbx3, panel K shows that the Myc-tagged deletion mutants can be IP’d by the anti-Myc antibody. In Panel K’, probing the anti-Myc IP for CAPERα shows that deletions more proximal than amino acid 655 disrupt the CAPERα/Tbx3 interaction.

The observation that C-terminal deletions interfere with CAPERα/Tbx3 led us to test whether the C-terminal repressor domain was important for the interaction (K, K’). In this case, because there was no interaction of the untagged ΔRD mutant with CAPERα, we thought it important to show that both the control and ΔRD mutant proteins were produced in the knockdown cells even though this has now been published.

In addition to revising the text and legend, we have modified the figure to illustrate which proteins are tagged and untagged in the schematic.

All the data are consistent with the conclusion that the C-terminal portion of Tbx3 and the C-terminal repressor domain are required for interaction with CAPERα.

*For example, in the experiments performed to determine the domain of TBX3 required for interaction with CAPERalpha. Use of shRNAs against TBX3 makes the interpretation of the results difficult, because it might also target some of the TBX3 deletion constructs. It would be preferable to perform these experiments using epitope-tagged TBX3 in the absence of shRNAs against TBX3*.

*Also, the rescue experiments using ectopic expression of CAPERalpha are difficult to interpret without knowing the region of CAPERalpha mRNA that is being targeted by the CAPERalpha shRNA. Please provide this information for all the shRNAs in the experimental methodology section. As in point 1 above, the authors should also include a second shRNA for their experiments to rule out shRNA-related “off-target” effects*.

These sequences were originally listed in the methods section but not specifically labeled as to which were employed, we have now clarified this and also state the region of the mRNA that is targeted by each shRNA in the Methods section.

*3) Statistical analyses are missing throughout the manuscript. Because of this it is not clear if some of the results presented in the manuscript are statistically significant. Please provide p-values for all of the results*.

We have now added p values for all data as requested. As is standard for the presentation of qPCR gene expression data, the data is presented in bar graph form with the mean and standard deviation represented as error bars. The p values are relative to control levels, as we are not directly comparing transcript levels TBX3 and CAPERα knockdown to one another; the point we are trying to convey with the expression data in Figure 2 is not that all senescence related genes are dysregulated in a statistically significant manner, but that the overall pattern of change seen in response to loss of TBX3 and CAPERα is similar (as was subsequently confirmed by the transcriptional profiling shown in Figure 7).

We have further clarified this point in the text.

The 3T5 growth curves shown in Figure 2 do not have error bars as each point on the population doubling curve represents the total cell count from each plate in an individual experiment. The curves shown are representative of results obtained from 2 independent experiments.

*4) The reviewers felt that the writing and presentation of the manuscript can be considerably improved. For example, there are uncommon usage of phrases (for example, embryonic mice, rather than mouse embryo; oncogene-induced stress physically dissociate, rather than disrupt; many others)*.

We have had the revised manuscript reviewed by additional readers and made edits requested by the reviewers and suggested by the readers.

*In general the figure legends are too sparse for the interested reader to fully understand how the experiments were carried out and presented.* eLife *does not have arbitrary word limits so there is no reason that additional details can't be provided. For example, in the mass spec experiment of*
Figure 1*, it is not clear which peaks are those of Caper and there is no information in the legends*.

We have expanded the legend for Figure 1 as requested; the spectrum fragmentation peaks are diagnostic for one of the six peptides that specifically identified CAPERα. The peptide sequence revealed by the mass spec trace shown is noted on the figure and is: C*PSIAAAIAAAVNALHGR.

We have also provided additional experimental details in all other figure legends as requested by the reviewers.

*In*
Figure 1
*and elsewhere there is no description as to what the black and red arrows indicate*.

This information was originally provided at the end of the figure legend, but we have now placed it at the beginning of the legend to alert the reader: “black arrowheads indicate IgG heavy chain and red indicate protein of interest (CAPERα or TBX3).”

*Finally, in the Method section, I could find no description of the methods used for the RIP experiments*.

We apologize for this accidental omission and have now provided the RIP methods.

*The following issues would be desirable to address for publication in* eLife*. We encourage the authors to address as many of these as possible*.

*1) The experiments in*
Figure 1
*E-G seem very tangential to the main conclusions of this study. The authors should consider removing these to help focus the manuscript*.

We respectfully disagree that the results are tangential as they illustrate key points that led us to pursue the interaction of these two proteins:

a) Endogenous *Caper*α is very broadly expressed whereas *Tbx3* expression is tissue specific and dynamic. Because we are interested in the developmental implications of this interaction as it may improve our understanding of the mechanisms underlying human ulnar-mammary syndrome associated with mutations in *TBX3*, it was important to determine *in vivo* co-localization in relevant tissues in developing embryos.

b) Since the data show that the proteins co-localize in the nuclei of some tissue types and not others (particularly apparent when comparing distal limb AER versus mesenchyme), they support the hypothesis that the interaction is regulated and tissue specific.

We have added this rationale to the text to allow the reader to appreciate our logic for these experiments.

*2) For all of the mRNA stability experiments can the authors provide a half-life estimate for the mRNA and discuss whether the half-life measurement is consistent with those obtained in other studies. Also, it would be desirable to confirm at least the key half-life experiments, such as those measuring p16 mRNA, by northern blot to rule out possible amplification artifacts that can arise by PCR*.

To obtain the estimated half-lives, we used linear regression on the data shown in Figure 6; those best fit lines, their equations and R values are now shown in Figure 6—figure supplement 1 and Figure 6—figure supplement 2, respectively. The differences in control half-lives between Figure 6 are likely attributable to the different control treatments: in A, control cells were transfected with pcDNA3.1 plasmid, while in B, control cells were transfected with control siRNA.

Note that in panel A, *p16*, *p14* and *E2F1* mRNAs do not decay in the time frame examined and their t_1/2_ are denoted as “n” (no decay).

The half-life of an mRNA is cell/context specific (as evident in the differences in control half-lives in A versus B) and in general, cell cycle regulatory genes have short half-lives [7]. The t_1/2_ of *p16* mRNA we observed in HFFs transfected with either control plasmid (t_1/2_ ∼3.9) or control siRNA (t_1/2_ ∼2.1) is similar to that reported in HeLa cells (t_1/2_ ∼2.9) [8]. The results we obtained were also similar to those reported for *MYC* mRNA [7, 9], *CDKN1A* mRNA in HT29-tsp53 cells [10] and ES cells [7], and *E2F1* mRNA in ES cells [7], The half- lives of *Rb* and *TGF*β*1* are mRNAs extremely variable and those we obtained in HFFs were shorter than reported in ES cells [7]. We have added this information to the Results section as requested.

We have also provided the p16 Northern blot (Figure 6—figure supplement 2), as requested. The results are consistent with those obtained by qPCR.

*3)*
Figure 6. *Can the increased hnRNPA1 binding to UAC1 following Ras expression be attributed to increased UAC1 RNA levels?*

Exactly: our data indicate that the increase in *UCA1* RNA levels in RAS cells (or after loss of TBX3 or CAPERα) allows *UCA1* to disrupt the *p16* mRNA /hnRNP A1 interaction leading to stabilization of *p16*.

4) Why is H3K27me3 levels measured in some but not all experiments?

We have added this data as requested.

*5)*
Figure 7*. In contrast to what is stated in the text, it is not clear that Caper levels markedly decrease in the cytoplasm*.

We agree and have removed that statement.

*Also in panels J and L the magnifications are not equivalent*.

The panels F-M are at the same magnification: cell and nuclear size of senescing cells is markedly increased (please see scale bar in G and also, Figure 2—figure supplement 3 where the same phenomenon occurs in *Tbx3* null MEFs).

*6) The authors use differential display to identify UAC1. This is an older and currently infrequently used procedure*. *Can the authors provide a rationale for their use of differential display?*

We appreciate the reviewers’ question: the rationale was that we had no funding for this line of investigation and Dr. Kumar already had the library of differential display primers and experience with this approach so at the time, this was the fastest and cheapest way for us to obtain insight into, and compare, what was happening after loss of TBX3 and CAPERα. Even though we knew it was not a comprehensive analysis, it revealed known and novel senescence effectors and because so little was known about *UCA1*, we pursued it.

*7) The authors show that CAPERalpha/TBX3 regulates histone modifications associated with p16 and UCA1 promoters. However, it is not clear how CAPERalpha/TBX3 achieves this. Are the changes in histone modifications a direct consequence of CAPERalpha/TBX3 activity? It would be desirable if the authors had results that address this issue but minimally the various possibilities should be discussed*.

TBX3 is known to interact directly with HDACs [11] but there are no reports of it, or CAPERα interacting with histone methyltransferases or demethylases. Our recently published Mass Spec screen for Tbx3/TBX3 interactors did not identify such factors however, the screen cannot be considered exhaustive as we did not reproducibly detect HDACs or transcription factors thought to interact with Tbx3. Future studies to specifically determine whether TBX3 and/or CAPERα interact with, recruit, or modify the function of EZH2, SUV39 and other methyltransferases will be informative. For the present manuscript, we have added additional discussion regarding known and potential roles for TBX3/CAPERα to regulate histone modifications.

*Additionally, it would be desirable for these experiments if total RNA polymerase II and elongating RNA polymerase II ChIP results could be included*.

We appreciate the reviewers’ point, but did not perform these ChIP experiments because we felt that the additional information to be gained was limited since the activating histone modifications observed correlated with the changes in gene expression.

*8) The results describing the role of CAPERalpha/TBX3 in premature senescence and oncogene-induced senescence are interesting but do not provide any insight into the role of this CAPERalpha/TBX3 pathway in either the process of aging or RAS-driven tumor initiation. Is there a tumor type where the authors can show this regulation occurs and modulates RAS-induced tumor initiation? This would add an important physiological context. Fibroblast-based experiments do not provide meaningful advances in our understanding for the role of CAPERalpha/TBX3 mediated senescence repression to either aging or tumorigenesis*.

We agree that the systems we employed (primary HFFs, mouse embryos and MEFs), while informative models, provide limited information that can be directly applied to aging or tumor initiation without extensive further experimentation. Based on the evidence suggesting that TBX3 influences the formation and progression of breast cancer, we performed a very preliminary assay on 8 human breast cancer tissue samples and 2 controls (see Figure 8, below). We assayed whether there was a correlation between the ability to detect CAPERα/TBX3 interaction by co-IP and the type of tissue (normal breast, Invasive ductal CA, Ductal Carcinoma in Situ, other carcinoma) or hormone receptor status. We did not observe any correlation however, these data are too preliminary to draw any conclusions because: 1) the sample size was small; 2) we are unable to perform immunohistochemistry on the available samples due to how they were preserved; 3) we do not have the ability to determine how heterogeneous the samples were. For these reasons, we have not included this data in the present manuscript. In the future, we hope to obtain funding for a study to examine CAPERα and TBX3 immunostaining and interaction in a larger clinical sample in which we have more detailed clinical information, and detailed tumor and control histo- and immune- pathology for each patient.Author response image 1.

Despite this uninformative preliminary experiment, our models, and the data derived therefrom, do support an important role for CAPERα and TBX3 in regulation of cell proliferation and senescence in developmental contexts. The role of the p16/RB pathway in tumor initiation is cell and context dependent and thus our observation that CAPERα/TBX3 complex regulates this pathway, combined with an increasing literature supporting roles for both TBX3 and CAPERα in multiple aspects of behavior of different tumor types indicate that several complex lines of investigation focused on specific tumor types will need to be pursued. We have added this to the Discussion.

*Also, it will be desirable to determine the state of telomerase expression/activity, telomere length and state of shelterin proteins in the experiments where premature senescence is measured*.

This manuscript addresses the effects of TBX3/CAPERα and UCA1 on non-telomere dependent mechanisms of senescence induction. Investigating whether TBX3 and/or CAPERα could additionally influence senescence by regulating telomere function is an interesting question, but it is beyond the scope of this paper.

*9) There are several previous reports that TBX3 represses cellular senescence. These previous should be discussed. Similarly, there are several reports on the role of long non-coding RNAs in the regulation of cellular senescence and these studies should also be discussed*.

In the original manuscript, we presented the fact that expression of *CDKN2A-p14*^*ARF*^ and *CDKN1A*-*p21*^*CIP*^ are repressed by TBX2 and TBX3 and that this mechanism has been proposed to account for the ability of overexpressed TBX2 and TBX3 to permit senescence bypass of *Bmi1*-/- and SV40 transformed mouse embryonic fibroblasts, respectively [12-14]_ENREF_7. Numerous overexpression studies have suggested a role for TBX3 in breast cancer initiation and progression (reviewed in [15, 16]) and recent papers have reported the tumorigenic and proinvasive effects of overexpressed TBX3 in melanoma cells [17, 18] which may in part result from repression of E-cadherin expression [19]. More relevant to our work on the importance of the CAPERα/TBX3 complex to prevent senescence and regulate cell proliferation are studies reporting that Tbx3 improves the pluripotency of iPS cells [20] and prevents differentiation of mouse ES cells [21]. As of this writing, we have found no other studies directed at the role of TBX3 in repressing senescence in primary cells or *in vivo* and this is one of the important contributions of our manuscript. We have added this information to the Discussion section as requested by the reviewers.

Regarding lncRNAs in senescence, although there has been a logarithmic increase in studies exploring their expression and function, lncRNA regulation of senescence remains largely unexplored. LncRNA *HOTAIR* has been shown to function as a scaffold to regulate ubiquitination of Ataxin-1 and Snurportin-1 to prevent premature senescence [22]. Global alterations in lncRNA expression have been reported in association with replicative senescence and telomere specific lncRNAs that regulate telomere during this process have been identified function [23, 24]. We found no reports of lncRNAs sufficient to induce senescence.

We have added the above information to the Discussion as requested by the reviewers.

References

1. Masutomi, K. et al. Telomerase maintains telomere structure in normal human cells. *Cell*
**114**, 241-53 (2003).

2. Boehm, J.S., Hession, M.T., Bulmer, S.E. & Hahn, W.C. Transformation of human and murine fibroblasts without viral oncoproteins. *Mol Cell Biol*
**25**, 6464-74 (2005).

3. Hong, H. et al. Suppression of induced pluripotent stem cell generation by the p53-p21 pathway. *Nature*
**460**, 1132-5 (2009).

4. Haga, K. et al. Efficient immortalization of primary human cells by p16INK4a-specific short hairpin RNA or Bmi-1, combined with introduction of hTERT. *Cancer Sci*
**98**, 147-54 (2007).

5. Elzi, D.J., Song, M., Hakala, K., Weintraub, S.T. & Shiio, Y. Wnt antagonist SFRP1 functions as a secreted mediator of senescence. *Mol Cell Biol*
**32**, 4388-99 (2012).

6. Kumar, P.P. et al. TBX3 Regulates Splicing In Vivo: A Novel Molecular Mechanism for Ulnar-Mammary Syndrome. *PLoS Genet*
**10**, e1004247 (2014).

7. Sharova, L.V. et al. Database for mRNA half-life of 19 977 genes obtained by DNA microarray analysis of pluripotent and differentiating mouse embryonic stem cells. *DNA Res*
**16**, 45-58 (2009).

8. Chang, N. et al. HuR uses AUF1 as a cofactor to promote p16INK4 mRNA decay. *Mol Cell Biol*
**30**, 3875-86 (2010).

9. Herrick, D.J. & Ross, J. The half-life of c-myc mRNA in growing and serum-stimulated cells: influence of the coding and 3' untranslated regions and role of ribosome translocation. *Mol Cell Biol*
**14**, 2119-28 (1994).

10. Melanson, B.D. et al. The role of mRNA decay in p53-induced gene expression. *RNA*
**17**, 2222-34 (2011).

11. Yarosh, W. et al. TBX3 is overexpressed in breast cancer and represses p14 ARF by interacting with histone deacetylases. *Cancer Res*
**68**, 693-9 (2008).

12. Jacobs, J.J. et al. Senescence bypass screen identifies TBX2, which represses Cdkn2a (p19(ARF)) and is amplified in a subset of human breast cancers. *Nat Genet*
**26**, 291-9 (2000).

13. Brummelkamp, T.R. et al. TBX-3, the gene mutated in Ulnar-Mammary Syndrome, is a negative regulator of p19ARF and inhibits senescence. *J Biol Chem*
**277**, 6567-72 (2002).

14. Prince, S., Carreira, S., Vance, K.W., Abrahams, A. & Goding, C.R. Tbx2 directly represses the expression of the p21(WAF1) cyclin-dependent kinase inhibitor. *Cancer Res*
**64**, 1669-74 (2004).

15. Liu, J. et al. TBX3 over-expression causes mammary gland hyperplasia and increases mammary stem-like cells in an inducible transgenic mouse model. *BMC Dev Biol*
**11**, 65 (2011).

16. Stephens, P.J. et al. The landscape of cancer genes and mutational processes in breast cancer. *Nature*
**486**, 400-4 (2012).

17. Peres, J. et al. The Highly Homologous T-Box Transcription Factors, TBX2 and TBX3, Have Distinct Roles in the Oncogenic Process. *Genes Cancer*
**1**, 272-82 (2010).

18. Peres, J. & Prince, S. The T-box transcription factor, TBX3, is sufficient to promote melanoma formation and invasion. *Mol Cancer*
**12**, 117 (2013).

19. Rodriguez, M., Aladowicz, E., Lanfrancone, L. & Goding, C.R. Tbx3 represses E-cadherin expression and enhances melanoma invasiveness. *Cancer Res*
**68**, 7872-81 (2008).

20. Han, J. et al. Tbx3 improves the germ-line competency of induced pluripotent stem cells. *Nature*
**463**, 1096-100 (2010).

21. Ivanova, N. et al. Dissecting self-renewal in stem cells with RNA interference. *Nature*
**442**, 533-8 (2006).

22. Yoon, J.H. et al. Scaffold function of long non-coding RNA HOTAIR in protein ubiquitination. *Nat Commun*
**4**, 2939 (2013).

23. Abdelmohsen, K. et al. Senescence-associated lncRNAs: senescence-associated long noncoding RNAs. *Aging Cell*
**12**, 890-900 (2013).

24. Yu, T.Y., Kao, Y.W. & Lin, J.J. Telomeric transcripts stimulate telomere recombination to suppress senescence in cells lacking telomerase. *Proc Natl Acad Sci U S A*
**111**, 3377-82 (2014).